# Comparing the clinical and economic efficiency of four natural surfactants in treating infants with respiratory distress syndrome

Reyhane Izadi[1], Payam Shojaei[2]*, Arash Haqbin[2], Abbas Habibolahi[3], Parvaneh Sadeghi-Moghaddam[4]

1 Department of Health Care Management, School of Management and Information Sciences, Shiraz University of Medical Sciences, Shiraz, Iran, 2 Department of Management, Shiraz University, Shiraz, Iran, 3 Neonatal Health Department, Ministry of Health and Medical Education, Tehran, Iran, 4 Neonatologist, Maternal Fetal and Neonatal Research Center, Tehran University of Medical Sciences, Tehran, Iran

* pshojaei@shirazu.ac.ir

**Data Availability Statement:** All relevant data are within the paper and its Supporting information files.

## Abstract

Surfactant therapy has revolutionized the treatment of respiratory distress syndrome (RDS) over the past few decades. Relying on a new method, the current research seeks to compare four common surfactants in the health market of Iran to determine the best surfactant according to the selected criteria. The research was a cross-sectional, retrospective study that used the data of 13,169 infants as recorded on the information system of the Iranian Ministry of Health. To rank the surfactants used, the following indicators were measured: re-dosing rate, average direct treatment cost, average length of stay, disease burden, need for invasive mechanical ventilation, survival at discharge, and medical referrals. The CRITIC (criteria importance through intercriteria correlation) method was used to determine the weight of the indicators, and MABAC (multi-attributive border approximation area comparison) was used to prioritize the surfactants. Based on the seven selected indicators in this research (re-dosing rate, average length of stay, direct medical cost per one prescription, medical referral rate, survival at discharge, disability-adjusted life years, number of newborns in need of invasive mechanical ventilation) and using multi-criteria analysis method, Alveofact was identified as the worst surfactant in infants with either more or less than 32 weeks' gestation. So that some criteria were worse in Alveofact group infants than other groups; for example, in the comparison of the Alveofact group with the average of the total population, it was found that the survival rate at discharge was 57.14% versus 66.43%, and the rate of re-dosing was 1.63 versus 1.39. BLES (bovine lipid extract surfactant) was the best alternative for infants more than 32 weeks' gestation, whereas Survanta was identified as best option for infants with less than 32 weeks' gestation. Curosurf showed an average level of functionality in the ranking. This study advises the policy makers in the field of neonatal health to increase the market share of more effective surfactants based on this study and other similar studies. On the other hand, neonatal health care providers are also advised to prioritize the use of more

**Funding:** The author(s) received no specific funding for this work.

effective surfactants if possible, depending on the clinical conditions and desired improvements.

**Competing interests:** The authors have declared that no competing interests exist.

## 1. Introduction

Respiratory distress is one of the most common causes of hospitalization in neonatal intensive care units (NICU). Reports suggest that 15% of full-term newborns and 29% of preterm newborns admitted to NICUs have acute respiratory complications [1]. The most prevalent types of respiratory distress are pneumonia, transient tachypnea of the newborn, meconium aspiration syndrome, and neonatal respiratory distress syndrome (RDS) [2].

Approximately 60% of infants with a gestational age of **less** than 30 weeks and 40% of infants weighing less than 1,500 g experience RDS [3]. RDS is a pulmonary disorder caused by alveolar surfactant deficiency, which can result in reduced alveolar surface tension, micro atelectasis, and decreased lung volume [4]. RDS often occurs due to defects in lung structure and an insufficient use of pulmonary surfactants [5, 6]. In addition, there are some RDS risk factors including diabetic mothers, preterm delivery, and the male gender [7]. Postnatal RDS involves symptoms such as blue skin and oral mucosa, rapid heartbeat, wheezing, chest indentation [8], pneumothorax, and intracranial hemorrhage [9]. In order to treat and prevent this disease, surfactant replacement therapy is used [10]

Numerous studies have compared the types of surfactants [11–13]; some have reported that different types of surfactants were almost equally effective [14], while others suggested significant differences between them [15]. Zayek et al. stated that due to the existence of several alternative products of pulmonary surfactant, they compared calfactant and poractant alfa with the aim of determining the strong pharmacological advantage in terms of cost and clinical indicators, and the results indicated the superiority of calfactant [16]. Although the effectiveness of surfactants in the prevention and treatment of RDS in infants has been well established, it is still unknown which surfactant is more effective [17]. On the other hand, scoring systems can be used to measure the performance of single therapeutic intervention over a time period, or used to compare the performance of one therapeutic intervention to others [18]. Scoring systems are used in all areas of medicine. Several parameters are evaluated and rated with points according to their value in order to simplify a complex clinical situation with a score [19]. The establishment of scoring system in medical areas is of great significance to effectively determine the severity of the disease, the rate of treatment success, and guide the treatment of doctors [20]. In this study, using selected performance indicators extracted from similar studies [16, 21–24] and a scoring-ranking system of medical interventions [25–29], the most effective surfactant in the treatment of RDS in infants has been determined. To this end, CRITIC method was adopted to calculate the weight of each indicator, and MABAC method was used in order to prioritize the surfactants. These two methods were adopted in this study due to the fact that CRITIC and MABAC were successfully combined in previous studies and made reliable results [25]. However, separately, CRITIC was applied in drug prioritization [26] and MABAC was used in several areas including medicine logistics management [27], healthcare sectors evaluation [28], supplier selection [29].

According to the unpublished statistics of the Ministry of Health and interviews with policy makers in the field of neonatal health, the prescription of surfactant in neonates has been growing rapidly in Iran in recent years, and on the other hand, surfactant is one of the vital and expensive drugs in Iran's pharmaceutical pharmacopoeia. Therefore, it is important to determine the type of superior surfactant that provides positive clinical results and is also

economically viable, as it can help specialists and decision makers to make appropriate decisions. In addition, although many studies have been conducted on various aspects of respiratory distress in neonates and the effect of different surfactants on it, most of the studies in this regard have only described and compared the results by different types of surfactants. This study was conducted with the aim of determining the best and worst surfactants in the treatment of respiratory distress syndrome in infants in Iran's health system. In this research, based on economic and clinical indicators, surfactants have been ranked for the first time with a new method.

## 2. Materials and methods

### 2.1 Research design and patient population

This study was a cross-sectional, retrospective research that tried to evaluate the effectiveness of four types of surfactants and rank them. The population under study included all infants with RDS who underwent surfactant therapy in Iran in 2018. This study is registered under the ethic approval code "IR.TUMS.IKHC.REC.1400.380" on 26/12/2021 by Tehran University of Medical Sciences, and all the methods used in the present study are in accordance with relevant guidelines and regulations.

The data of these infants were extracted using the census method, in fact, sampling was not done, and all eligible infants were included in the study using the census method. The entry and exit criteria were as follows:

- Inclusion criteria: All infants with RDS in Iran who had undergone surfactant therapy;

- Exclusion criteria: The following cases were excluded from the study: Infants with RDS who were discharged with the consent of their parents (or legal guardians) before full recovery, those who were not primarily diagnosed with RDS, and those with an unidentified gestational age. Also, infants who were referred to another center before surfactant administration were excluded from the study.

The research data were extracted from the Iranian Maternal and Neonatal Network (IMAN net), as per the necessary permits and with the support of the Iranian Ministry of Health. It should be noted that the design of this national network is such that it is mandatory to record information for many variables, so the missing data in this study is minimal. Because the outcomes of surfactant injection could be different among different age groups [21], the data were analyzed separately for two groups of infants: those with a gestational age more than 32 weeks and those with a gestational age less than 32 weeks.

To ensure that there is no significant difference in the baseline in the neonates of the four surfactant groups, four indicators of gestational age, birth weight, Apgar 1st minute and Apgar 5th minute were used, which indicate the general health status of the newborns at birth and before surfactant administration; And it was determined in advance that before the administration of surfactant, there was no statistically significant difference in the health status of infants in different surfactant groups (see some other variables in Table S3.1 in S3 File).

### 2.2 Study variables and measured outcomes

The purpose of this study was to rank four natural surfactants used in Iran (Alveofact, Survanta, Curosurf and BLES) in the RDS treatment process. The indicators investigated in this research were extracted from the literature on this topic. The indicators that were included in the main analyzes of the article were previously determined to be statistically significantly different between the four groups of infants. Although there were some other relevant indicators,

they were removed from the analysis due to the lack of statistically significant differences (see Table S3.1 in S3 File). The selected indicators are as follows:

1. Redosing rate: A surfactant would be most efficacious when it required less redosing;

2. Average length of stay (ALOS): This indicator referred to the length of hospital stay;

3. Average direct cost of treatment: For each prescription, two types of cost were computed: fixed cost and variable cost. Fixed cost included consumables (chip tubes, NG tubes, etc.), which were the same for all prescriptions. Variable cost included the price of the surfactant vial in 2018, which varied according to the type of surfactant. By combining these two types of cost, the direct cost of treatment was determined in each case the surfactant was administered. Following that, given the number of doses for each infant, the average direct cost of treatment for each infant was determined. Finally, the average direct cost of treatment (as well as the total direct cost of treatment) was determined based on the type of surfactant used;

4. Medical referral rate (severity of illness): In the Iranian health system, newborns receive surfactant only at specialized hospitals that have an NICU. According to interviews with the directors of the Neonatal Health Department (affiliated with the Ministry of Health) and the unpublished data from the Ministry, referrals after surfactant administration often occur due to clinical reasons and the severity of patients' conditions. Therefore, the high rate of referrals in relation to a particular type of surfactant would indicate that the surfactant was not sufficiently effective. As a result of the lack of improvement or worsening of the health condition of the babies after receiving surfactant, these babies are referred to the specialized hospitals that have more advanced equipment, more facilities, more specialized manpower and more up-to-date technologies. In Iran's health system, these hospitals are defined regionally and based on geographical proximity.

5. Survival at discharge: A discharge order (issued by a doctor) represents the most ideal outcome for a newborn given his/her good health condition. A surfactant type that could lead to the ordered discharge of a greater percentage of newborns could be more effective in treating RDS. This criterion would help to determine the chances of the baby surviving after the administration of the surfactant in question. The term "survival" in this criterion was used to describe patients who were alive at the time of discharge from the hospital [30].

6. Disability-adjusted life years (DALY) (per 1,000 neonates with RDS): DALY measured the gap between the current state and the ideal state, or the state in which all people would live up to standard age in perfect health. According to life tables on the World Bank website, the standard life expectancy at birth in Iran is 75.40 years for men and 77.66 for women. This index, like other indicators, was calculated separately for all the four groups of surfactants under investigation. The DALY index, which is very important in prioritizing health interventions, was a combination of life time with disability and time lost due to premature death, calculated as follows:

$$DALY = YLL + YLD$$

where:
YLL = years of life lost due to premature mortality.
YLD = years lived with disability.
In this study, this index was calculated by the method of the World Health Organization, taking into account the discounting rate of three percent and unequal age weighting [31].

The YLL formula is as follows:

$$YLL = (N/r) * (1 - e^{-rL})$$

where:

N = number of deaths.

L = standard life expectancy at age of death (years).

r = discount rate (discount rate of 0.03).

In this formula N is considered as the number of neonatal deaths in four surfactant groups. The formula of YLD is as follows:

$$YLD = (I \times DW \times L \, (1 - e^{-rL}))/r$$

where:

I = number of incident cases (-).

DW = disability weight (-).

L = duration of disability (years).

r = discount rate

e = The number e, also known as Euler's number, is a mathematical constant that is approximately equal to 2.71828.

In this formula, I is the number of newborns in all four surfactant groups and L is ALOS. According to the WHO disability weight scale, the DW for lower respiratory tract infections is 0.28 [31]. Because RDS is a lower respiratory tract infection [32], in this study DW was considered to be 0.28.

7. Number of newborns in need of invasive mechanical ventilation: A large number of newborns in need of invasive mechanical ventilation or a long period of invasive mechanical ventilation could reflect the low efficiency of an injected surfactant.

## 2.3 Data analysis

The data were primarily classified and analyzed in Excel and SPSS through items of descriptive statistics such as frequency, percentage, mean, and standard deviation. Following that, the CRITIC method was used to determine the weights of the indicators, while the MABAC method helped to rank the surfactants; Microsoft Excel was used to apply both analysis methods.

**2.3.1 CRITIC method.** The CRITIC method was initially developed in 1995 by Diakoulaki et al. as a technique for calculating the weight of indicators in multi-criteria decision-making problems. In this method, the opinion of experts is not important and the relative weight of indicators is determined by correlation coefficients and standard deviation of data [33]. According to Diakoulaki et al. (1995), the steps of CRITIC method are as follows:

Step 1: Similar to other multi-criteria decision-making methods, a decision matrix containing n indicators and m alternatives is developed in the first step. The decision matrix is then normalized using Eq (1) where xij is the value of each element of matrix, x min and x max are the minimum and maximum values of the matrix in each column, respectively.

$$r_{ij} = \frac{xij - x\,min}{x\,max - x\,min} \tag{1}$$

Step 2: In the next step, the value of c for each column of the decision matrix is calculated using Eq (2). rij represents elements of the normalized decision matrix, σ is the standard deviation of the data of each column and m is the number of indicators.

$$c = \sigma \sum_{i=1}^{m} (1 - \mathrm{rij}) \tag{2}$$

Step 3: In the final step, the value of c obtained for each column of the matrix is divided by the sum of the values of c, in order to obtain the final weight of each indicator, which is shown in Eq (3).

$$w = \frac{c}{\sum_{i=1}^{m} c} \tag{3}$$

**2.3.2 MABAC method.** The MABAC method is a recently developed multi-criteria decision-making technique used to rank alternatives in multi-criteria decision-making models. The basis of the MABAC method originated from the definition of the distance of the indicator function of each alternative from the border approximation area. MABAC was developed by Pamučar & Ćirović (2015).[34]The steps of the MABAC method are presented as follows:

***Step 1***: The initial decision matrix—including indicators and alternatives—is formed, where $x_{ij}$ is the value of alternative $i$ ($i = 1, \ldots, m$) according to indicator $j$ ($j = 1, \ldots, n$).

***Step 2***: The initial decision matrix is normalized according to Eqs (4) and (5) for the positive and negative indicators, respectively.

$$n_{ij} = \frac{x_{ij} - \min\left(x_{ij}\right)}{\max\left(x_{ij}\right) - \min\left(x_{ij}\right)}, \ \textit{for benefit} \tag{4}$$

$$n_{ij} = \frac{\max\left(x_{ij}\right) - x_{ij}}{\max\left(x_{ij}\right) - \min\left(x_{ij}\right)}, \ \textit{for cost} \tag{5}$$

***Step 3***: Calculating the weighted normalized matrix (V). The elements of matrix V are determined based on Eq (6):

$$V_{ij} = w_j\left(n_{ij} + 1\right) \tag{6}$$

where $n_{ij}$ is the elements of normalized matrix and $w_j$ is the weight of indicator $j$.

***Step 4***: Calculating the border approximation area (BAA) matrix. The BAA for all indicators (G) is determined based on Eq (7):

$$g_j = \left(\prod_{i=1}^{m} V_{ij}\right)^{\frac{1}{m}} \tag{7}$$

Matrix G is formed as follows:

$$G = \begin{bmatrix} g_1 & \cdots & g_n \\ \vdots & \ddots & \vdots \\ g_1 & \cdots & g_n \end{bmatrix}$$

where $n$ is the number of indicators.

**Step 5**: Calculating the distance of alternative $i$ from the BAA. This distance ($q_{ij}$) is determined as the difference between the elements of matrix V and matrix G (Eq (8)).

$$Q = V - G = \begin{pmatrix} v_{11} & \cdots & v_{1n} \\ \vdots & \ddots & \vdots \\ v_{m1} & \cdots & v_{mn} \end{pmatrix} - \begin{bmatrix} g_1 & \cdots & g_n \\ \vdots & \ddots & \vdots \\ g_1 & \cdots & g_n \end{bmatrix}$$
$$= \begin{pmatrix} q_{11} & \cdots & q_{1n} \\ \vdots & \ddots & \vdots \\ q_{m1} & \cdots & q_{mn} \end{pmatrix} \tag{8}$$

If $q_{ij} > 0$, alternative A is close to the ideal solution. If $q_{ij} < 0$, alternative A is close to the anti-ideal solution.

**Step 6**: Prioritizing the alternatives. The sum of the distance of the alternatives from the BAA (the elements of matrix Q) is calculated based on Eq (9). The higher this value for each alternative $i$, the higher the ranking of that alternative.

$$S_i = \sum_{j=1}^{n} q_{ij}, \; i = 1, \ldots, m \tag{9}$$

## 3. Results

### 3.1 Patient demographics and clinical characteristics

According to the inclusion and exclusion criteria of the study, the number of infants studied in this research decreased from 16551 to 13169 cases. The number of 1133 babies due to not having RDS problem, 1093 babies due to voluntary discharge, 1100 babies due to uncertain gestational age, and finally 56 cases due to referral to another hospital before surfactant administration, all these babies (3382 cases) were excluded from the study. The descriptive statistics of the study showed that 42.05% of the newborn infants were girls, the mean gestational age was 32.24 (±4.55) weeks, and the mean birth weight was 1876.94 (±813) grams. More than 74% of the infants were born by Cesarean section and the mean Apgar scores of the first and fifth minutes were 6.94 and 8.31, respectively. Table 1 shows the descriptive statistics of the surfactant groups.

### 3.2 Description of criteria

The measurement of the outcomes after surfactant injection revealed that, except for ALOS, other outcomes were significantly different in relation to the types of surfactants. It

**Table 1. Patient characteristics of the overall study population and surfactant cohorts.**

| Variables | Overall Study Population (N = 13169) | Alveofact (N = 308) | BLES (N = 2124) | Curosurf (N = 7788) | Survanta (N = 2949) |
|---|---|---|---|---|---|
| **Gender** | | | | | |
| Female | 5533(42.05) | 131(42.53) | 876(41.24) | 3167(40.66) | 1359(46.08) |
| **Gestational age** | | | | | |
| Mean, wk [a] | 32.24 (±4.55) | 32.19 (±4.41) | 32.19 (±4.77) | 32.27 (±4.58) | 32.44 (±4.44) |
| ≤ 32 weeks | 7227(54.87) | 177(57.46) | 1213(57.10) | 4389(56.35) | 1448(49.10) |
| **Birth weight** | | | | | |
| Mean, gr [b] | 1876.94 (±813) | 1856.42 (±715) | 1836.06 (±852) | 1906.37 (±839) | 1908.91 (±849) |
| ≤ 1500gr | 5376(40.82) | 124(40.25) | 976(45.95) | 3275(42.05) | 1465(49.67) |
| **Delivery method** | | | | | |
| Cesarean | 9842(74.73) | 233(75.64) | 1567(73.77) | 5752(73.85) | 2290(77.65) |
| **Apgar Score** | | | | | |
| First minute Apgar [c] | 6.94 | 6.94 | 6.94 | 6.93 | 6.95 |
| Five-minute Apgar [d] | 8.31 | 8.28 | 8.33 | 8.33 | 8.32 |

*Data are presented as number (%)

[1] Vaginal birth after cesarean

P-value:

[a] = 0.07 /

[b] = 0.09/

[c] = 0.18 /

[d] = 0.13

was found that ALOS was 13.71 days. The average direct treatment cost was $398.49. Alveofact was the most expensive type and required the largest amount of re-dosing. The survival rate was 66.43%, as 26.10% of the infants died and 7.45% needed referrals due to the severity of their condition. The DALY index showed that 14820.53 days of healthy living were lost due to NRDS. The outcomes of administrating the surfactant types are shown in Table 2.

**Table 2. Clinical and economic outcome for overall study population and surfactant cohorts.**

| Variables | Overall study population | Alveofact | BLES | Curosurf | Survanta |
|---|---|---|---|---|---|
| Average length of stay, days | 13.71 | 11.90 | 13.28 | 15.86 | 13.80 |
| Average direct medical cost | 398.49 | 627.94 | 258.35 | 385.61 | 322.05 |
| Re-dosing rate | 1.39 | 1.63 | 1.36 | 1.30 | 1.30 |
| Needing invasive mechanical ventilation | 5862(44.51) | 163(52.92) | 838(39.45) | 3550(45.58) | 1311(44.45) |
| Medical referrals rate | 1012 (7.68) | 32(10.38) | 275(12.94) | 447(5.73) | 228(7.73) |
| Survival at discharge | 8749(66.43) | 176(57.14) | 1272(59.88) | 5258(67.51) | 2043(69.27) |
| Mortality rate, n | 3438(26.10) | 100(32.46) | 577(27.16) | 2083(26.74) | 678(22.99) |
| DALY, day (year) | 14820.53(40.6041) | 17909.77(49.0681) | 14927.34(40.8968) | 14983.65(41.0510) | 13939.82(38.1912) |
| YLL, day (year) | 14799.98(40.5478) | 17893.47(49.0232) | 14909.05(40.8467) | 14961.81(40.9912) | 13920.79(38.1391) |
| YLD, day (year) | 20.57(0.0563) | 16.41(0.0449) | 18.32(0.0501) | 21.86(0.0598) | 19.03(0.0521) |

Amount (percentage)

## 3.3 The decision matrix

The research decision matrix consisted of seven indicators for surfactant evaluation and four different types of surfactants as the alternatives. The seven relevant indicators were the re-dosing rate (I1), the average length of stay (I2), the average direct treatment cost (I3), medical referral rate (I4), survival at discharge (I5), and DALY per 1,000 infants (I6), and the number of infants in need of invasive mechanical ventilation (I7). In addition, the alternatives in the decision matrix were four different types of surfactant: Alveofact, BLES, Curosurf, and Survanta. Table 3 shows the decision matrix of the infants.

As mentioned earlier, the weights of the indicators were measured through the CRITIC method by calculating the correlation coefficients and standard deviation of the data (see Table 4).

According to the results of the CRITIC method presented in Table 4, the most important indicator in evaluating the surfactants in both infants with a gestational age of more and less than 32 weeks was the "re-dosing rate" (I1). In addition, the least important indicator in the evaluation of the surfactant types for infants with a gestational age less than 32 weeks was the "average length of stay" (I2). The least important indicator for infants with a gestational age more than 32 weeks was the "medical referral rate" (I4).

The resultant weights were used to rank the surfactants through the MABAC method. As mentioned above, following the steps of the MABAC method, the decision matrix was first normalized based on Eqs (4) and (5). It should be noted that all the indicators of the decision matrix, except for "survival at discharge" (I5), had a negative nature. After multiplying the weights obtained via the CRITIC method in the normalized decision matrix, the matrices of $V_{ij}$ and G were calculated according to Eqs (6) and (7). The Q matrix, then, was calculated through difference between the elements of matrix V and matrix G (Eq (8)). Ultimately, the final $S_i$ values were calculated based on the Eq (9). Table 5 shows these values; it should be noted that higher $S_i$ values would highlight the higher quality of an alternative.

According to the results in Table 5, the best surfactant for infants with a gestational age less than 32 weeks was Survanta, followed by BLES and Curosurf. Alveofact was the last surfactant. In contrast, for infants with a gestational age more than 32 weeks, the best surfactant was BLES and the worst one was Alveofact.

**Table 3. Infant decision matrix.**

| Type of surfactant | ≤32 | | | | | | | >32 | | | | | | |
|---|---|---|---|---|---|---|---|---|---|---|---|---|---|---|
| | I1 | I2 | I3 | I4 | I5 | I6 | I7 | I1 | I2 | I3 | I4 | I5 | I6 | I7 |
| Alveofact | 1.75 | 13.91 | 675.7071 | 20 | 73 | 14238.2263 | 116 | 1.5 | 9.89 | 580.1830 | 12 | 103 | 3671.5466 | 47 |
| BLES | 1.40 | 17.27 | 266.2549 | 136 | 580 | 12289.5255 | 610 | 1.31 | 9.30 | 250.4575 | 139 | 692 | 2637.8212 | 228 |
| Curosurf | 1.35 | 21.28 | 402.1170 | 275 | 2404 | 11689.0680 | 2313 | 1.24 | 10.44 | 369.1220 | 172 | 2854 | 3294.5916 | 1237 |
| Survanta | 1.35 | 17.35 | 335.4233 | 117 | 794 | 11120.0987 | 754 | 1.24 | 10.24 | 308.6833 | 111 | 1249 | 2819.7182 | 557 |

**Table 4. Indicator weights.**

| Gestational age | I1 | I2 | I3 | I4 | I5 | I6 | I7 |
|---|---|---|---|---|---|---|---|
| ≤32 | 0.1868 | 0.1143 | 0.1473 | 0.1200 | 0.1438 | 0.1429 | 0.1446 |
| >32 | 0.1883 | 0.1060 | 0.1568 | 0.0999 | 0.1487 | 0.1445 | 0.1555 |

**Table 5.** The values of $S_i$.

| Type of surfactant | $S_i$ (>32) | Rank | $S_i$ (= <32) | Rank |
|---|---|---|---|---|
| Alveofact | -0.183981929 | 4 | -0.152136722 | 4 |
| BLES | 0.238804661 | 1 | 0.140056915 | 2 |
| Curosurf | -0.000511128 | 3 | 0.01485928 | 3 |
| Survanta | 0.153497584 | 2 | 0.203576162 | 1 |

### 3.4 Validation of results and sensitivity analysis

In order to examine the obtained results using MABAC, the all type of surfactants will be ranked using two other methods: MAIRCA (multi-attributive ideal-real comparative analysis) method, [35]and VIKOR (VIseKriterijumska Optimizacija I Kompromisno Resenje) method [36]. The result of ranking the surfactants with these methods is shown in Table 6. As can be seen, there is very little difference between the rankings.

The difference is only in rank 1 and 2. In this way, it has been proven that the results using the MABAC methods do not deviate from the results obtained using other methods. Although it was obvious that the deviations were not significant, the results were verified by applying Spearman's correlation coefficient (SCC) [37]. The SCC values are given in Table 7.

As can be seen from Table 7, the SCC values range from 0.8 to 1. This presents a very high rank correlation value. Accordingly, it can be concluded that the results of the MABAC method are satisfactory, respectively, the robustness of the presented method has been proven.

## 4. Discussion

The purpose of this study was to compare the performance of four surfactants based on seven selected indicators to determine the best surfactants, relying on the multi-criteria decision making method (MDCM) in Iran. Detailed discussions are provided below.

### 4.1 Comparative evaluation: Survanta and BLES as the best, Corsurf and Alveofact with weaker performance

The findings of the study showed that in both groups of infants with a gestational age less and more than 32 weeks, Alveofact was the least preferable type, while Curosurf had an average level of functionality. It was also found that Survanta was the best choice for infants with a gestational age less than 32 weeks, while BLES was the best surfactant for infants with a gestational age more and more than 32 weeks. The two types were identified as the best surfactants.

**Survanta.** Reporting results consistent with those of the present study, Mussavi et al. revealed that in treating RDS through Survanta replacement therapy was more effective than administrating Alveofact [21]. In another similar study, Hammoud et al. compared the efficacy

**Table 6.** Validation of results and sensitivity analysis.

| Type of surfactant | VIKOR | | VIKOR | | MAIRCA | | MAIRCA | |
|---|---|---|---|---|---|---|---|---|
| | Q (>32) | Rank | Q (< = 32) | Rank | Q (>32) | Rank | Q (< = 32) | Rank |
| Alveofact | 1 | 4 | 1 | 4 | 0.1733 | 4 | 0.1552 | 4 |
| BLES | 0.1457 | 2 | 0.1647 | 2 | 0.0676 | 1 | 0.0822 | 2 |
| Curosurf | 0.6206 | 3 | 0.5239 | 3 | 0.1274 | 3 | 0.1135 | 3 |
| Survanta | 0.1008 | 1 | 0 | 1 | 0.0889 | 2 | 0.0663 | 1 |

**Table 7. SCC values for alternative ranks obtained by different methods.**

|  | VIKOR (<32) | MABAC (<32) | MAIRCA (<32) |  | VIKOR (>32) | MABAC (>32) | MAIRCA (>32) |
|---|---|---|---|---|---|---|---|
| VIKOR (<32) | 1 |  |  | VIKOR (>32) | 1 |  |  |
| MABAC (<32) | 1 | 1 |  | MABAC (>32) | 0.8 | 1 |  |
| MAIRCA (<32) | 1 | 1 | 1 | MAIRCA (>32) | 0.8 | 1 | 1 |

of Alveofact and Survanta in terms of illness severity and mortality. They also confirmed that neonates who received Survanta experienced fewer side-effects compared to those who received Alveofact. As such, Survanta was more effective [38].

**BLES.** In a randomized controlled trial (RCT) by Lemyre et al., the efficacy and safety of BLES and Curosurf were compared. The results showed that although there was no significant difference in primary outcomes (extubation, bronchopulmonary dysplasia, etc.), in secondary outcomes Curosurf was associated with higher mortality and BLES with higher probability of survival. Therefore, BLES was recognized as the more effective surfactant [39]. Sarokolai et al. compared the adverse effects of BLES and Curosurf, observing that although the two surfactants had almost the same treatment efficacy, although BLES was a generally more reliable product for surfactant therapy [40]. Other studies, in line with the findings of the present research, confirmed that Survanta [21, 38] and BLES [39, 40] can perform better than other types of surfactants in some indicators.

**Curosurf.** The observations of Najafian et al. were in line with the results of this study. They found that the neonates with RDS who were of gestational age of 32 weeks, Curosurf showed less efficiency and less safety than Survanta and had more side-effects following its injection [3]. In a retrospective cohort study, Paul et al. stated that Curosurf had no superiority over Survanta in the treatment of RDS [41]. In another similar study by Baroutis et al., contrary to the findings of the present study, it was found that Alveofact and Curosurf had better functionality than Survanta [42]. Because more than half of the newborn infants in this study were less than 32 weeks of gestational age, the findings of Najafian et al. [3] were more relevant to those of this study.

**Alveofact.** Proquitté et al. measured clinical outcomes in neonates with RDS who were treated with Alveofact and Curosurf. It was found that there was no significant difference in clinical efficacy between the two groups [43]. In another study, Yalaz et al. compared the efficacy of two natural exogenous surfactants, namely Alveofact and Survanta, in the treatment of RDS. They showed no statistically significant differences in the final effects and adverse effects between the two groups [44]. Mussavi et al. compared the efficacy of Survanta and Alveofact in the surfactant treatment process of premature neonates with RDS. They found that some clinical variables were worse in certain age groups who received Alveofact [21].

## 4.2 Comparing the indicators based on surfactant types

### 4.2.1 Re-dosing rate (I1)

The findings of this study showed that Alveofact required significantly higher amounts of re-dosing than the other three surfactants. Meanwhile, Survanta and Curosurf did not differ much in terms of this index. The findings of Mussavi et al., consistent with those of the present study, showed that the average number of surfactant injections among patients receiving Alveofact was significantly more than those who received Curosurf and Survanta, and that these two surfactants were not significantly different from each other [21]. Manizheh et al. also showed that surfactant re-dosing among neonates who received Alveofact was significantly

more than those who were treated with Curosurf [22]. Contrary to the results of the present study, Fox et al. [45] and Mirzarahimi et al. [23] showed that Curosurf reduced the need for re-dosing compared to other surfactants. However, the findings observed by Mussavi et al. [21] were consistent with those of the present research, because their study was specifically similar with the present research in terms of its context (Iran), research sample, and the drugs investigated.

**4.2.2 Average length of stay (I2).** The highest ALOS index was found in the case of Curosurf, while the lowest value of this index was seen in the case of Alveofact. Corroborating the results of the present study, Manizheh et al. conducted a comparative study of preterm neonates with RDS and showed that ALOS was higher in the group using Curosurf than the one receiving Alveofact [22]. However, in another similar study, it was found that ALOS was higher in the group treated with Alveofact than in those receiving Survanta and Curosurf [21]. Because more than half of the newborns in this study were less than 32 weeks of gestational ag, the findings of Manizheh et al [22], who focused on preterm neonates, were more relevant to observations of this study.

**4.2.3 Direct medical cost (I3).** The results of this study showed that BLES and Survanta, as the superior surfactants, imposed significantly lower costs on the health system, compared to Alveofact and Curosurf. Sarokolai et al. examined the cost index in preterm neonates with RDS to determine the most efficacious type of surfactant, showing that this index was significantly lower in the group treated with BLES than the one receiving Curosurf. As such, in 91.50% of infants who received Curosurf, cost exceeded $200 (inflation-adjusted cost = $ 273.48), although that cost was only observed in case of 8.50% of infants who used BLES [40]. In a retrospective cohort study of infants at risk of RDS to compare the efficacy and safety of Calfactant and Curosurf, Zayek et al. found that the cost of Curosurf-based treatment per patient was $ 1,160.62 (inflation-adjusted cost: $1229.56), which was 38% higher than the cost imposed by using Calfactant ($838.34 (inflation-adjusted cost: $877.08)) [16]. Marsh et al conducted a study to compare the pharmacoeconomic profiles of Survanta and Curosurf via a cost-minimization analysis. These analyses would suggest Curosurf may offer a less costly, clinically-equivalent option. Different treatment models using Curosurf (compared to Survanta) resulted in cost savings ranging from 53% ($949.67 (inflation-adjusted cost: $1296.21)) to 20% ($180 (inflation-adjusted cost: $245.68)) [46]. In a study aimed at evaluating economic and therapeutic efficiency (based on drug therapy cost index, duration of respiratory support, duration of hospitalization, side effects, etc.), Brown et al. reported higher average medication costs ($1756.44 vs. $1329.78 (inflation-adjusted cost: $1860.77 vs. $1408.77)) for Poractant Alfa (Curosurf) compared with Beractant (Survanta). While many clinical indicators were not significantly different between the groups [47]. A cost-effectiveness analysis study of surfactant therapy in the treatment of NRDS showed that the use of Poractant Alfa (Curosurf) is a superior option compared to Beractant (Survanta). Cost-effectiveness ratio was 4585 ($5067.57) per saved life for Poractant Alfa and 5087 ($5590.35) per saved life for Beractant [48]. Another study aimed at comparing the efficacy and safety of bovine lung phospholipid and Poractant Alfa injection in the treatment of neonatal hyaline membrane disease showed that treatment costs in the for Poractant Alfa group were significantly lower than the bovine lung phospholipid group [49]. Note: all compared studies were adjusted for dollar currency and inflation (2018).

**4.2.4 Medical referral rate/severity of illness (I4).** The results of this study showed that, in general, infants who were forced to refer to other medical centers due to the deterioration of their clinical conditions had the highest and lowest number of referrals in the Alveofact and Curosurf groups, respectively. In a randomized controlled trial (RCT) study, pneumothorax in infants with a gestational age less than 32 weeks and PDA in infants with a gestational age

more than 32 weeks were more likely in groups who received Alveofact, compared with those who received Curosurf and Survanta [21]. Najafian et al. observed similar rates of complication including sepsis, pneumonia, necrotizing enteric colitis (NEC), intraventricular hemorrhage (IVH), and retinopathy of prematurity (ROP) among groups who were treated with Curosurf and Survanta [3]. Given the framework of this study, the specific effects of surfactants cannot be stated with certainty. But in general, based on the results of this study and the available evidence [3, 21, 46], it can be stated that probably the severity of the illness in the case of Curosurf and Survanta was better than Alveofact and BLES.

**4.2.5 Survival at discharge (I5).** Investigating the neonatal outcome showed that those who were treated with Survanta, the best surfactant, showed the highest survival rate, which was the highest rate of discharge by a physician's order and the lowest mortality rate. Evidence has shown that surfactant replacement therapy in neonates with RDS, regardless of the type of surfactant, increases the likelihood of survival [24]. The results of a clinical meta-analysis showed that natural surfactants significantly reduced mortality compared to artificial ones [50]. Among natural surfactants, however, Fujii et al. reported that survival was more likely in neonates receiving Curosurf [51]. Yet, in line with the results of this study, the findings of Bloom et al. confirmed that Survanta was more effective in increasing survival rates. They compared Survanta and BLES in a multicenter clinical trial, observing that in preventive surfactant prescriptions the probability of survival to discharge in neonates under 600 g was 74% among those who received Survanta and was 37% among those who received BLES. The groups, of course, were significantly different (P = 0.007) [24]. In general, it is clear that the use of surfactant increases the survival rate [24], and among the types of surfactants, animal-derived products are more effective than artificial ones [50]. Finally, among the types of animal surfactants, the results of this research, along with others [52], showed that Survanta was more efficient in increasing the survival rate or the survival at discharge.

**4.2.6 Mortality rate/DALY(I6).** It was found that mortality in infants using Survanta (the most effective surfactant) was significantly lower, and thus in this group, compared to the other three surfactants, a lower disease burden was imposed on the community. It was also found that a large part of DALY was related to YLL. Further evidence confirmed that in general about 93% of DALY was associated with YLL [53].

Evidence has shown that pharmaceutical innovation can leave different impacts on YLL, suggesting that some pharmaceutical classes are more successful [54]. As the results of this study clarified, Survanta was more effective and had the lowest YLL value. Frank et al. conducted an econometric study of the effect of pharmaceuticals on DALY and its two components, namely YLL and YLD. They showed that drugs launched between 1986 and 2001 reduced DALY by 2.31 million in 2016 [53]. In a study by Bloom et al., it was found that in prophylactic surfactant administration the group receiving Survanta had significantly less mortality than the one receiving BLES [52]. Comparing Survanta and Curosurf, Bozdag et al. found that in the case of pulmonary hemorrhage-related mortality, the two types were not significantly different [46].

Considering the results of this study, along with available evidence [52, 55], mortality was lower in groups that received Survanta, compared to other surfactants. Even studies that found results contrary to this observation [13, 46] did not usually report statistically significant differences. In general, it was found that Survanta, as the most effective surfactant, had the lowest mortality rate and the lowest YLL and DALY values. As further evidence suggests [54], drugs can affect the burden of disease in the community differently; among the four surfactants investigated in this study, Survanta mostly reduced the burden of RDS in Iran.

**4.2.7 Invasive mechanical ventilation (I7).** Analyzing this index showed that the infants who received the top two surfactants, Survanta and BLES, significantly needed less invasive

ventilation. Another study that found similar results revealed that the need for invasive mechanical ventilation support in infants with a gestational age of more than 32 weeks who received Survanta was significantly lower than those who received Curosurf and Alveofact [21]. Mirzarahimi et al. found that the mean duration of ventilation was significantly shorter in infants treated with Survanta than those receiving Curosurf [23]. Najafian et al. reported a similar rate of need for continuous positive nasal air way pressure in groups treated with Curosurf and Survanta [3]. In general, studies that explored the need for invasive mechanical ventilation observed that Survanta was superior to other surfactant types in terms of the need for invasive mechanical ventilation [3, 21] or the length of the invasive mechanical ventilation period [23].

As with all retrospective studies, one of the inherent limitations of this study was the heterogeneity of the patients in the groups under investigation. Also, although the specialized research team tried to focus on the IMAN net data that reported "the severity of illness" as the main cause of referrals after surfactant administration, some referrals were caused by non-clinical and unexplained reasons that could not be perfectly categorized. These limitations, of course, were negligible in the entire research population. In addition, another limitation of this research is that although according to the evidence, the need for mechanical ventilation and its duration is one of the indications of the efficacy of surfactant therapy, it is possible that factors other than RDS caused it. In this research, we were limited to the data extracted from the Iranian Maternal and Neonatal Network (IMAN net), and it was not possible to comprehensively examine all the factors. Another main limitation of this research was the lack of proportion in the number of prescriptions in four types of surfactants. In fact, in Iran's health system, the use of surfactant (and of course many other drugs) is not optional and depends on the political and economic conditions of the country. The specific political conditions of Iran, the presence of extensive sanctions and the sharp drop in currency value in different periods of time, lead to the non-existence of a certain type of surfactant or the presence of only a certain type of surfactant in the health market. It should be noted that according to the "National Guidelines for Surfactant Therapy in Neonates with Respiratory Distress", there is no recommendation to use or not use a specific type of surfactant in a specific group of infants. In general, according to the opinion of decision makers and policy makers in the field of newborn health in Iran, the most important issue affecting the choice of the type of surfactant is the country's political-economic conditions, which affects the existence of various surfactants.

## 5. Conclusion

The results showed that BLES was the best alternative for infants with a gestational age more than 32 weeks, whereas Survanta was the best option for infants with a gestational age of less than 32 weeks. Alveofact was the worst option for treating RDS. The results of this study were meant to help healthcare professionals and policymakers to make informed decisions and should not be used as a substitute for professional medical advice. Neonatal health policymakers are advised to focus on increasing the market share of highly effective surfactants. Future studies could rank a combination of artificial and natural surfactants and use more indicators to evaluate their functionality.

## Supporting information

**S1 File. How to calculate MABAC and CRITIC methods in neonates $\leq = 32$.** (DOCX)

**S2 File. How to calculate MABAC and CRITIC methods in neonates >32.**
(DOCX)

**S3 File. Baseline characteristics, abnormalities, and risk factors.**
(DOCX)

**S4 File. STROBE statement—Checklist of items that should be included in reports of *cross-sectional studies*.**
(DOC)

**S5 File.**
(XLSX)

## Acknowledgments

The authors thank the neonatal department of the ministry of health, for providing statistical data from the Iranian Maternal and Neonatal Network (IMAN Net) to support the research.

## Author Contributions

**Conceptualization:** Reyhane Izadi.

**Data curation:** Reyhane Izadi.

**Investigation:** Abbas Habibolahi, Parvaneh Sadeghi-Moghaddam.

**Methodology:** Payam Shojaei, Arash Haqbin.

**Software:** Payam Shojaei.

**Supervision:** Payam Shojaei, Abbas Habibolahi, Parvaneh Sadeghi-Moghaddam.

**Writing – original draft:** Reyhane Izadi.

**Writing – review & editing:** Parvaneh Sadeghi-Moghaddam.

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
