## [Decision Letter · Decision Letter 0]

6 Mar 2023

PONE-D-23-01167Comparing the Clinical and Economic Efficiency of Four Natural Surfactants in Treating Infants with Respiratory Distress SyndromePLOS ONE

Dear Dr. Shojaei,

Thank you for submitting your manuscript to PLOS ONE. After careful consideration, we feel that it has merit but does not fully meet PLOS ONE’s publication criteria as it currently stands. Therefore, we invite you to submit a revised version of the manuscript that addresses the points raised during the review process.

ACADEMIC EDITOR: Dear Authors, please find the comment of both reviewers and answered and amend your manuscript accordingly to their suggestions and remarks

We look forward to receiving your revised manuscript.

Kind regards,

Stefan Grosek, Ph.D., M.D.,

Academic Editor

PLOS ONE

Journal Requirements:

a) Did participants provide their written or verbal informed consent to participate in this study?

Additional Editor Comments:

Dear Authors

The theme of this manuscript is very interesting for neonatologists but the the whole manuscript has many ambiguities and inconsistencies addressed by the reviewers and myself too. I hope that after major revision this manuscript could be eligible for publication.

Reviewers' comments:

Reviewer's Responses to Questions

**Comments to the Author**

1. Is the manuscript technically sound, and do the data support the conclusions?

Reviewer #1: No

Reviewer #2: Yes

2. Has the statistical analysis been performed appropriately and rigorously? 

Reviewer #1: I Don't Know

Reviewer #2: I Don't Know

3. Have the authors made all data underlying the findings in their manuscript fully available?

Reviewer #1: Yes

Reviewer #2: Yes

4. Is the manuscript presented in an intelligible fashion and written in standard English?

Reviewer #1: Yes

Reviewer #2: Yes

5. Review Comments to the Author

Reviewer #1: The submitted study is a study of high importance in neonatology

However, it has methodological problems:

- What are the clinical criteria chosen to redose surfactant adminsitration? Are they always the same between units? Do they vary according to gestational age?

- Among the judgement criteria used is the length of hospital stay. However, this factor is very dependent on the gestational age of birth which does not seem to be taken into account in the calculation.

-atuhros say that reffering to a NICU may depend on the severity of the patient. However, clinical severity can have some influence in the need of surfactant redosing. It seems that this factor is not considered

Minor remarks

the end of the first paragraph is confusing with RDS describing different lung distress

the table 1 could be simplified (one line at a time) female are not male , <1500g is the opposite of >1500g ...

Reviewer #2: I appreciate authors who have brought into the light about unattended facet of newborn care.

There should be a STROBE checklist attached with (manuscript) to address all the aspects of observational studies.

Please find comments in word file. needs major revision.

6. PLOS authors have the option to publish the peer review history of their article (what does this mean?). If published, this will include your full peer review and any attached files.

Reviewer #1: No

Reviewer #2: **Yes: **anurag fursule

---

## [Author Response · Author response to Decision Letter 0]

24 May 2023

PONE-D-23-01167

Comparing the Clinical and Economic Efficiency of Four Natural Surfactants in Treating Infants with Respiratory Distress Syndrome

PLOS ONE

Thank you for submitting your manuscript to PLOS ONE. After careful consideration, we feel that it has merit but does not fully meet PLOS ONE’s publication criteria as it currently stands. Therefore, we invite you to submit a revised version of the manuscript that addresses the points raised during the review process.

Authors’ response: Thank you for your positive feedback. Based on this email from you, we tried to fully implement the PLOS ONE's publication criteria.

ACADEMIC EDITOR: 

• Dear Authors, please find the comment of both reviewers and answered and amend your manuscript accordingly to their suggestions and remarks

Authors’ response: We carefully reviewed all the points raised by dear reviewers and responded to their suggestions and remarks. They gave us very valuable and useful comments. If they had a recommendation, we implemented it. If they wanted corrections, we did it. If there was any ambiguity for them, we answered it and attached additional information to clear the ambiguity. We tried to give full and comprehensive answers to all their comments. We spent a lot of time doing these revisions. However, if there are any other corrections or any new suggestions that the dear editors and respected reviewers consider necessary, please let us know about it, we will do it if possible.

Authors’ response: Thank you for your guidance. We sent the following items when submitting our revised manuscript:

1. “Response to Reviewers”

2. “Revised Manuscript with Track Changes”: In this file, we have marked the changes made in the original manuscript with yellow highlighter.

3. “Manuscript”

4. S1_File

5. S2_File

6. S3_File

7. S4_File

8. S5_File

9. Research Ethics Committees Certificate

10. Iran's national guideline in surfactant therapy for neonates

Authors’ response: No thanks, we don't want to make any changes. As we said earlier, this research was not financially supported by any institution.

Authors’ response: This research is not a laboratory work and no laboratory protocol was used in it.

We look forward to receiving your revised manuscript.

Kind regards,

Stefan Grosek, Ph.D., M.D.,

Academic Editor

PLOS ONE

Journal Requirements:

Authors’ response: Thank you for this comment and for sending us these two links. We revised the manuscript point by point based on these two files. We tried to implement all the mentioned points, if there is a problem, please give us feedback.

a) Did participants provide their written or verbal informed consent to participate in this study?

Authors’ response: In this research, the data recorded in the national system have been used (Iranian Maternal and Neonatal Network (IMAN net)). In the data file used, the identity of the patients was hidden, no written or verbal consent was taken from anyone. This research is a retrospective study and the analyzed data was related to the whole country (about 16 thousand babies), and it was practically impossible to obtain consent. Access to the data was done with the official permission of the Ministry of Health of Iran. This research is approved by the ethics committee of Tehran University of Medical Sciences. In the method section, we stated this issue as follows:

"This study is registered under the ethic approval code “IR.TUMS.IKHC.REC.1400.380” on 26/12/2021 by Tehran University of Medical Sciences, and all the methods used in the present study are in accordance with relevant guidelines and regulations."

We also attached the relevant certificate under the title "Research Ethics Committees Certificate", please check it.

Authors’ response: The raw and preliminary data related to this research, which was extracted from the national system of Iran, is confidential and cannot be sent. However, the underlying data used to reach the conclusions presented in the manuscript can be submitted. We attached this data file in supporting information files with the title "S5_File". This data file is sufficient to support the correct implementation of research analytical techniques. If more data is needed, please let us know. Although this dataset (S5_File) and the information presented in S1_File and S2_File cover all the analyzes and data presented in the manuscript, we may be able to provide you with more data (if approved by the Neonatal Health Department of the Ministry of Health).

Additional Editor Comments:

Dear Authors

The theme of this manuscript is very interesting for neonatologists but the the whole manuscript has many ambiguities and inconsistencies addressed by the reviewers and myself too. I hope that after major revision this manuscript could be eligible for publication. 

Authors’ response: We appreciate the effort and time you have taken to provide feedback on our manuscript. We have carefully reviewed all concerns and have done our best to address each one, and we hope that our edits and responses satisfactorily address the issues and concerns raised. Your comments were very valuable and useful, on the other hand, we also spent a lot of time and effort to respond to your suggestions and corrections. In order to track recommended and performed corrections, please note the following color scheme: The green highlights show the reviewers' comments, the yellow highlights show revisions within the manuscript, and gray highlights indicate the authors' response to the reviewers' and editors' comments. And finally, although we have done our best to make corrections and recommendations according to the comments of the editor and reviewers, if you think more corrections are needed, we would like to know your opinions so that we can implement your suggestions if possible.

 

Reviewers' comments:

Reviewer's Responses to Questions

Comments to the Author

1. Is the manuscript technically sound, and do the data support the conclusions?

Reviewer #1: No

Reviewer #2: Yes

Authors’ response: Thank you for your positive and negative feedback. In response to this comment, we attached two files S1_File and S2_File. In these two files, we have presented the main analysis (MABAC and CRITIC methods) and the process of extracting the results from the available data in full, to give you the assurance that the data technically supports the conclusion. You can check the technical analysis used in this research step by step, to ensure that the tests were done accurately. 

It should be noted that we did not have sampling and used the census. Although this issue was stated in the text of the manuscript, in response to this comment, we added the following text to the manuscript:

"The data of these infants were extracted using the census method, in fact, sampling was not done, and all eligible infants were included in the study using the census method."

If you have specific suggestions and recommendations regarding this question (Is the manuscript technically sound, and do the data support the conclusions?), please give us feedback. If it is possible for us, we will definitely do it.

2. Has the statistical analysis been performed appropriately and rigorously?

Reviewer #1: I Don't Know

Reviewer #2: I Don't Know

Authors’ response: Thank you for your honest response. The statistical analyzes used in this research are few and simple. The main analysis of this research is based on multi-criteria decision making methods. One of the strengths of this research is the use of this analysis method. We used the CRITIC method to determine the weight of the indicators and the MABAC method to prioritize the alternatives (four types of surfactants). To better understand these two methods, check the two files S1_File and S2_File. In these two, we have explained the analysis method completely and step by step.

Also, to understand the nature of the CRITIC method, you can read the article by Diakoulaki et al. [1]. And for a better understanding of the MABAC method, if you wish, read the article by Pamučar & Ćirović [2]. Please ask us if you have any questions about these analysis methods.

3. Have the authors made all data underlying the findings in their manuscript fully available?

Reviewer #1: Yes

Reviewer #2: Yes

Authors’ response: Thank you for your comment.

4. Is the manuscript presented in an intelligible fashion and written in standard English?

Reviewer #1: Yes

Reviewer #2: Yes

Authors’ response: Thank you for your positive comment.

 

5. Review Comments to the Author

Reviewer #1: The submitted study is a study of high importance in neonatology 

However, it has methodological problems:

Authors’ response: We appreciate the effort and time you have taken to provide feedback on our manuscript. We have carefully reviewed all concerns and have done our best to address each one, and we hope that our edits and responses satisfactorily address the issues and concerns raised. Your comments were very valuable and useful, on the other hand, we also spent a lot of time and effort to respond to your suggestions and corrections. In order to track recommended and performed corrections, please note the following color scheme: The green highlights show the reviewers' comments, the yellow highlights show revisions within the manuscript, and gray highlights indicate the authors' response to the reviewers' and editors' comments. And finally, although we have done our best to make corrections and recommendations according to the comments of the editor and reviewers, if you think more corrections are needed, we would like to know your opinions so that we can implement your suggestions if possible.

- What are the clinical criteria chosen to redose surfactant adminsitration? Are they always the same between units? Do they vary according to gestational age? 

Authors’ response: Surfactant therapy for newborns in Iran is done according to the "National Guideline for Surfactant Prescription in Neonates". In this guideline, the re-dosing indicators are exactly the same for all four types of surfactants. In fact, in one section of this guideline, three indicators are written for re-dosing in all types of surfactants; and gestational age is not mentioned in any of these three indicators (Note: Because the gestational age in the four groups of infants examined was not statistically different, even if the re-dosing depended on the gestational age... it still did not affect the results of this research.). The document "National Guideline for Surfactant Prescription in Neonates" was published by the Iranian Ministry of Health in Persian language, and unfortunately, this document does not have an English version at all. Following your comment, we have attached the original version of this guideline, which is in Farsi. About this guideline, we must say that "European Consensus Guidelines on the Management of Respiratory Distress Syndrome" was one of its main references. It should be noted that although the main basis for compiling Iran's national guideline was the " European Consensus Guidelines...", other references (63 references) were also used in compiling this national guideline, and this guideline has been localized in all aspects to the conditions of Iran. In response to your opinion, we tried to briefly translate the three re-dosing indicators written in the Iranian National Guidelines into English, which are presented below:

Indications for repeated use of the drug:

- To repeat the prescription, make a decision based on the baby's need for oxygen and his/ her clinical condition.

- Within 48 hours after the first administration of surfactant, if one of the following conditions exists, the indication is to repeat the administration of surfactant.

o After 6 to 12 hours, the baby still has a tracheal tube and needs a mean airway pressure (MAP) of more than 7 cm H2O and oxygen of more than 40%.

o With CPAP, with a minimum H2Opressure of 6-7 cm, there is a need for more than 40% oxygen.

o Chest X-ray should be done before administering repeated doses.

- Among the judgement criteria used is the length of hospital stay. However, this factor is very dependent on the gestational age of birth which does not seem to be taken into account in the calculation.

Authors’ response: Thank you for your review and comment. As we have written in Table 1 of the manuscript, the gestational age of the infants in the four investigated groups (according to the four types of surfactants examined) was not significantly different. Therefore, although the variable of gestational age can potentially affect the duration of hospitalization, in this study, since the variable of gestational age does not have a significant difference in the four groups, it cannot create a significant difference in the duration of hospitalization between the four groups under investigation. In other words, although the gestational age had an effect on the length of hospitalization, the extent of this effect was similar between the four investigated groups and did not create a significant difference between these four groups in terms of length of hospitalization). Therefore, the difference in the duration of hospitalization between the four investigated groups is probably for a reason other than the gestational age, because the gestational age of all four groups is similar.

In addition, the main goal of the research is to investigate the clinical-economic effects of various surfactants. For this purpose, variables that were statistically significantly different were included in MABAC and CRITIC analyses. The variable of gestational age was not significantly different between the infants in the four investigated groups. We looked for indicators that were significantly different between the four groups of infants and then included them in MABAC and CRITIC analysis. In fact, the variables that were included in the multi-criteria decision analysis (MABAC and CRITIC method) were definitely statistically significantly different between the four groups of infants. And about other variables that were not significantly different between groups of infants, these variables were never included in the main analyzes of the manuscript at all.

Thank you again for your feedback. If you have any special suggestion about this comment, please share it with us.

-atuhros say that reffering to a NICU may depend on the severity of the patient. However, clinical severity can have some influence in the need of surfactant redosing. It seems that this factor is not considered.

Authors’ response: Thank you very much for taking the time to review this manuscript. I'm sorry, I think that maybe I didn't understand your meaning from this comment correctly and accurately. I will answer you based on my current understanding of this comment. If there is any ambiguity or we have not answered your question correctly, please tell us more clearly what you mean.

What we meant by "referring to a NICU may depend on the severity of the patient" was actually the severity of the disease after surfactant administration. Based on your comment, we corrected this sentence as follows:

"Also, although the specialized research team tried to focus on the IMAN net data that reported “the severity of illness” as the main cause of referrals after surfactant administration, some referrals were caused by non-clinical and unexplained reasons that could not be perfectly categorized."

As shown in Table 1 of the manuscript, infants in the four surfactant groups do not differ significantly based on baseline characteristics. In Table 1 of the manuscript, we only stated five very common characteristics that were reported by other similar studies, to avoid data overload in the manuscript. While the research team, from the beginning of this research, based on the comprehensive data set that was extracted from the Iranian Maternal and Neonatal Network (IMAN net), examined the groups of infants in terms of many characteristics that their data is available in the system, and it was found that there is no significant difference in these characteristics. But due to the large number of these variables, it is challenging to present them in the text of the manuscript, express the results and discuss them; and the research team prefers to report only common variables used in other similar studies (the variables in Table 1 of the manuscript are taken from similar studies; with the aim of showing the baseline parameters indicating the general state of health between the four investigated groups, in which there was no significant difference in infants). In fact, with the aim of ensuring that the babies of the four groups were similar before the administration of surfactant, we examined a large number of variables that directly and indirectly affected the subject of the research, and finally it was found that most of these variables had no significant difference between the four groups (variables related to the clinical conditions of the newborn, risk factors related to the mother, and diseases and abnormalities of newborns, etc.).

Consider the severity of the disease in two situations, before and after the administration of surfactant. 

Before the administration of surfactant, some key indicators that influence the severity of the disease (gestational age, birth weight), or can potentially indicate the severity of the disease (Apgar score (min 1 and 5), were not significantly different between the four groups of infants (a large number of other variables affecting the severity of the disease are written in Table 1 of the manuscript and Table S3.1 in S3_File, please check them.). So, in short....

.

.

All infants were similar in terms of baseline parameters affecting disease severity

.

.

On the other hand, all these babies had RDS.

.

.

 All of them received surfactant.

.

.

And only some of them were referred after receiving surfactant...

.

.

Therefore, it can be concluded that because the infants of the four groups did not differ significantly in many factors affecting the severity of the disease or the indicators showing the severity of the disease before the administration of surfactant, but the amount of their referral due to the severity of the disease after the administration of surfactant was significantly different... So, depending on the type of surfactant, probably the prescribed surfactant was not as effective as it should be, and as a result, the severity of the respiratory disease increased (or recovery did not occur); In other words, it is likely that the lack of effect of the first dose depending on the type of surfactant (and not the severity of the disease) has led to the repetition of subsequent doses.

In addition, it should be noted that some of these referrals are for reasons other than the severity of the disease, which we have already mentioned in the limitations of the research. We stated that some of these referrals may be due to reasons other than the severity of the disease, which is not possible for us to investigate further. This is as follows:

"….., some referrals were caused by non-clinical and unexplained reasons that could not be perfectly categorized. These limitations, of course, were negligible in the entire research population."

Thank you again for carefully reviewing this manuscript. If you have any suggestion in this regard, please share it with us.

Minor remarks

the end of the first paragraph is confusing with RDS describing different lung distress

Authors’ response: Thank you for your valuable comment. There was a typo at the end of the first paragraph, thank you for giving us feedback. We have edited this sentence as follows:

“The most prevalent types of respiratory distress are pneumonia, transient tachypnea of the newborn, meconium aspiration syndrome, and neonatal respiratory distress syndrome (RDS) [2].”

Minor remarks

the table 1 could be simplified (one line at a time) female are not male , <1500g is the opposite of >1500g ...

Authors’ response: Thank you for this comment. According to your comment, we corrected and simplified Table 1. Please check it in the manuscript. 

It should be noted that some authors did not agree with this edit in Table 1, but in the end, because the information in this table is completely written in In Table S3 of S3_File in Appendices, we made this edit according to your comment. Please check it in the manuscript.________________________________________

Reviewer #2: I appreciate authors who have brought into the light about unattended facet of newborn care.

There should be a STROBE checklist attached with (manuscript) to address all the aspects of observational studies.

Please find comments in word file. needs major revision.

Authors’ response: We appreciate the effort and time you have taken to provide feedback on our manuscript. We have carefully reviewed all concerns and have done our best to address each one, and we hope that our edits and responses satisfactorily address the issues and concerns raised. Your comments were very valuable and useful, on the other hand, we also spent a lot of time and effort to respond to your suggestions and corrections. In order to track recommended and performed corrections, please note the following color scheme: The green highlights show the reviewers' comments, the yellow highlights show revisions within the manuscript, and gray highlights indicate the authors' response to the reviewers' and editors' comments. And finally, although we have done our best to make corrections and recommendations according to the comments of the editor and reviewers, if you think more corrections are needed, we would like to know your opinions so that we can implement your suggestions if possible. Please check our responses and revisions to the comments raised in the Word file on the following pages.

Also, thank you for suggesting that we complete the STROBE checklist. Based on this comment, we completed the STROBE checklist. If corrections were needed based on this checklist, we made them and marked these revisions in the manuscript with yellow highlights. We have attached this checklist under the title "S4_File", please check it.

6. PLOS authors have the option to publish the peer review history of their article (what does this mean?). If published, this will include your full peer review and any attached files.

Do you want your identity to be public for this peer review? For information about this choice, including consent withdrawal, please see our Privacy Policy.

Reviewer #1: No

Reviewer #2: Yes: anurag fursule

Authors’ response: Dear Dr. Anurag Fursule and dear reviewer#1, we sincerely appreciate you. You gave us very valuable and useful comments. If you have a recommendation, we implemented it; If you wanted corrections, we did it; If there was any ambiguity, we answered it and attached additional data to clear the ambiguity. We tried to give a complete and comprehensive answer to all your comments, however, tell us any other corrections you think are necessary. Also, if you have any suggestions, let us know and we will do it if possible.

 

The comments raised in the Word file by Reviewer 2:

I appreciate authors who have brought into the light about unattended facet of newborn care.

We appreciate the effort and time you have taken to provide feedback on our manuscript. We have carefully reviewed all concerns and have done our best to address each one, and we hope that our edits and responses satisfactorily address the issues and concerns raised. Your comments were very valuable and useful, on the other hand, we also spent a lot of time and effort to respond to your suggestions and corrections. In order to track recommended and performed corrections, please note the following color scheme: The green highlights show the reviewers' comments, the yellow highlights show revisions within the manuscript, and gray highlights indicate the authors' response to the reviewers' and editors' comments. And finally, although we have done our best to make corrections and recommendations according to the comments of the editor and reviewers, if you think more corrections are needed, we would like to know your opinions so that we can implement your suggestions if possible.

There should be a STROBE checklist attached with (manuscript) to address all the aspects of observational studies.

Authors' response: Thank you for suggesting that we complete the STROBE checklist. Based on this comment, we completed the STROBE checklist. If corrections were needed based on this checklist, we made them and marked these revisions in the manuscript with yellow highlights. We have attached this checklist under the title "S4_File", please check it.

Abstract

1. CRITIC/MABAC/BLES acronym should be expanded

Authors' response: Thanks for reminding us of this. According to your comment, these abbreviations were explained in the abstract. Please check these in the manuscript. These revisions are as follows:

MABAC (multi-attributive border approximation area comparison)

CRITIC (criteria importance through intercriteria correlation)

BLES (bovine lipid extract surfactant)

2. The standard terminology for Live discharge rate should be survival at discharge.

Authors' response: You suggested a very suitable term, thank you very much. We reviewed the entire text of the manuscript and replaced the term "survival at discharge" with "live discharge rate" and "survival to hospital discharge rate". You can check these changes in the text of the manuscript, we corrected it in 8 places of the article. The places in the manuscript where this editing was done are as follows: In the method and results of the abstract, in part "2.2 Study Variables and Measured Outcomes" in the method section of the manuscript, in Table 2, twice in the section "3.3 The Decision Matrix", twice in the section "4.2.5 Survival at discharge (I5)".

3. Some scores denoting the inferiority of Alveofactant should be mentioned in the abstract. The display of numbers will amplify the effect on reader about overall results of study.

Authors’ response: Thank you for your suggestion. According to your comment, we added some of these scores, which indicated that Alveofact surfactant was inferior, in the results section of the abstract. This revision is as follows:

...", Alveofact was identified as the worst surfactant in infants with either more or less than 32 weeks’ gestation. So that some criteria were worse in Alveofact group infants than other groups; for example, in the comparison of the Alveofact group with the average of the total population, it was found that the survival rate at discharge was 57.14% versus 66.43%, and the rate of re-dosing was 1.63 versus 1.39."

Introduction

1. It is very long. RDS is well studied entity so the history and type of surfactants can be ignored.

Authors’ response: Thank you very much for your suggestion. According to your comment, the history and type of surfactants were removed from the introduction section, and it was only stated in one sentence that surfactants are used to treat RDS. The red sentences were removed, and the yellow highlighted sentence was added. These revisions are as follows:

o “In order to treat and prevent this disease, surfactant replacement therapy is used [11]."

o Fujiwara, a Japanese scientist, conducted a LANDMARK study in 1980, and for the first time in the history of the disease treated ten premature babies with RDS using artificial surfactants [10]. Surfactant replacement therapy has revolutionized the treatment of neonatal respiratory failure in recent decades. Surfactants are used to treat and prevent RDS, although there are debates about the symptoms that would justify surfactant administration. There are currently various artificial and natural surfactants available in commercial health markets worldwide [11].

o Evidence suggests natural surfactants are more effective [12].

o In the Iranian health market, there are four natural surfactants available: Beractant (Survanta), Bovactant (Alveofact), Poractant Alpha (Curosurf), BLES. 

o Curosurf is derived from minced porcine lungs, whereas the other three types are bovine extracts [13].

2. The utility of functionality scores should be discussed in modern medicine

Authors’ response: Thank you very much for your valuable suggestion. According to your comment, an explanation about scoring in medicine was added to the introduction section. You can check it in the third paragraph of the introduction. These revisions are as follows:

“Although the effectiveness of surfactants in the prevention and treatment of RDS in infants has been well established, it is still unknown which surfactant is more effective [17]. On the other hand, scoring systems can be used to measure the performance of single therapeutic intervention over a time period, or used to compare the performance of one therapeutic intervention to others [18]. Scoring systems are used in all areas of medicine. Several parameters are evaluated and rated with points according to their value in order to simplify a complex clinical situation with a score [19]. The establishment of scoring system in medical areas is of great significance to effectively determine the severity of the disease, the rate of treatment success, and guide the treatment of doctors [20]. In this study, using selected performance indicators extracted from similar studies [16, 21-24] and a scoring-ranking system of medical interventions [25-29], the most effective surfactant in the treatment of RDS in infants has been determined. To this end, CRITIC method was adopted to calculate the weight of each indicator, and MABAC method was used in order to prioritize the surfactants. These two methods were adopted in this study due to the fact that CRITIC and MABAC were successfully combined in previous studies and made reliable results [25].”

Methods

1. “A surfactant would be most functional when it required less redosing;”- I would rather use word efficacious than functional in this sentence.

Authors’ response: Thank you for your suggestion. According to your comment, this sentence was edited as follows:

“A surfactant would be most efficacious when it required less redosing;”

2. Medical referral rate: where are babies referred after surfactant administration.?

Authors’ response: The process of surfactant therapy for newborns is only and necessarily performed in the NICU. The equipment and the level of expertise of human resources in the NICU of different hospitals in Iran can be very different depending on the region. In all NICUs, surfactant injection is possible. If the clinical condition of the patient (neonate) worsens after the injection of surfactant (or does not improve) and the patient needs a neonatal subspecialty and more specialized care and more advanced equipment, it will inevitably be referred to another hospital.

In fact, the decision to refer a patient to a center with a better NICU happens in two situations: (1) Or the patient is referred before surfactant administration; (2) Or the patient is referred after surfactant administration.

Babies of group (1) were not included in the study at all, because the reason for the referral of these babies could be anything other than the effect of surfactant. We explained this issue in the manuscript exit criteria section. This is as follows:

"Also, infants who were referred to another center before surfactant administration were excluded from the study."

And the neonates of group (2), they have been referred to centers with more equipped NICUs, with more specialized manpower and more advanced facilities (sub-specialized hospitals at a higher referral level) than the hospital where they were born. In response to your comment, we have added additional explanations in this regard to the manuscript, which are as follows:

"As a result of the lack of improvement or worsening of the health condition of the babies after receiving surfactant, these babies are referred to the super specialty hospitals that have more advanced equipment, more facilities, more specialized manpower and more up-to-date technologies. In Iran's health system, these hospitals are defined regionally and based on geographical proximity."

Based on the interview we had with the directors of the Department of Neonatal Health of the Ministry of Health of Iran: They stated that most of the babies in this group (group (2)) had to be referred due to the lack of improvement in their health condition and even worsening of their condition. On the other hand, considering that the babies in the four surfactant groups did not differ significantly in many demographic characteristics, baseline parameters, clinical conditions, diseases and abnormalities, etc. (Please see Table 1 in the manuscript and Table S3.1 in S3_File), all of them had RDS, received surfactant, and were referred after that.... therefore, it can be concluded that probably the prescribed surfactant was not as effective as it should be, and as a result, the severity of the respiratory disease increased. Other similar studies with similar target population and research objectives used this index like ours.

However, there are other potential reasons for referral (considering the above explanations and the homogeneity of the babies in the four groups, this type of referral is few). Based on the interview we had about this comment with experts in this field in the Ministry of Health, it was stated that the percentage of this type of referrals in the entire target population of this research is very small and can be ignored. We have previously mentioned this as one of the limitations of the research. It is as follows:

"Also, although the specialized research team tried to focus on the IMAN net data that reported “the severity of illness” as the main cause of referrals, some referrals were caused by non-clinical and unexplained reasons that could not be perfectly categorized. These limitations, of course, were negligible in the entire research population."

Based on the data set that the research team has and has done the analysis, it is possible but very difficult to separate referrals by reason. If the dear reviewer insists on separating the types of referrals, we will do so, although our prediction is that the results will not change because referral for non-clinical reasons are very few. Please let us know your decision on this matter.

I think that maybe I did not understand your meaning from this question exactly. If I didn't answer your question correctly, please give us feedback more clearly what is your purpose of this question. Thank you again for the time you spent on this manuscript and the feedback you gave us. If you think we need to make corrections or revisions, we will gladly accept them if possible.

3. Sample size calculation for study?

Authors’ response: Dear reviewer, the entire research population (i.e. All infants with RDS in Iran who had undergone surfactant therapy) were included in the study by census method, which included 16,551 cases; And only those who met the exclusion criteria were excluded from the study, as a result of which the research population decreased from 16,551 to 13,169. In fact, all eligible babies were included in the research with the census method. These explanations about the size of the studied population are presented in the section "2.1 Research Design and Patient Population". And the number of research population is stated in the first paragraph of the results section. However, in response to your comment, we emphatically stated that sampling was not done and all eligible infants were included in the study. Thank you again for your valuable comments. The mentioned revisions were added to the "2.1 Research Design and Patient Population " section, which are as follows:

“The data of these infants were extracted using the census method, in fact, sampling was not done, and all eligible infants were included in the study using the census method."

4. How were MABAC and CRITIC computed? Any software?

Authors’ response: Excel software was used to perform MABAC and CRITIC calculations. In line with your comment, a sentence was added to the relevant paragraph as follows:

“The data were primarily classified and analyzed in Excel and SPSS through items of descriptive statistics such as frequency, percentage, mean, and standard deviation. Following that, the CRITIC method was used to determine the weights of the indicators, while the MABAC method helped to rank the surfactants; Microsoft Excel was used to apply both analysis methods. "

Authors’ response: Regarding how to calculate MABAC and CRITIC, we have tried to provide brief and comprehensive explanations in the manuscript, which you can see in the "2.3.1 CRITIC" and "2.3.2 MABAC" sections. Also, in order to avoid additional writing in the manuscript, we mentioned Diakoulaki (1995) reference about CRITIC method and Pamučar & Ćirović (2015) reference about MABAC method, which readers can refer to if they want to know more specialized information. However, in response to your comment about how to calculate these two methods, we have attached a Word file under the title "S1_ File and S2_File". In this file, the calculation steps of both methods are written in detail separately for babies less than and more than 32 weeks of gestational age. We thank you again for this comment, surely this attached file will provide a better understanding for the readers.

5. MABAC and CRITIC background calculation and description are not need?

Authors’ response: In response to your comments, a file titled " S1_File & S2_File" has been attached, which includes calculations related to MABAC and CRITIC methods. please check it. In addition, we have written the background of these specialized techniques (MABAC and CRITIC) in the manuscript, which you can check. These are as follows:

2.3.1 CRITIC method

The CRITIC method was initially developed in 1995 by Diakoulaki et al. as a technique for calculating the weight of indicators in multi-criteria decision-making problems. In this method, the opinion of experts is not important and the relative weight of indicators is determined by correlation coefficients and standard deviation of data [33]. According to Diakoulaki et al. (1995), the steps of CRITIC method are as follows:

2.3.2 MABAC method

The MABAC method is a recently developed multi-criteria decision-making technique used to rank alternatives in multi-criteria decision-making models. The basis of the MABAC method originated from the definition of the distance of the indicator function of each alternative from the border approximation area. MABAC was developed by Pamučar & Ćirović (2015) [34]. The steps of the MABAC method are presented as follows:

Results

1. Gestational age and Birth weight should be mentioned as mean (standard deviation)

Authors’ response: Thank you very much for your valuable comment. According to your suggestion, we wrote the mean (standard deviation) for birth weight and gestational age. We made these corrections in part "3.1 Patient Demographics and Clinical Characteristics" of the Results section in Table 1 and the paragraph above it. You can check these corrections in the manuscript. These corrections are as follows:

Gestational age: Mean, wk a 32.24 (±4.55) 32.19 (±4.41) 32.19 (±4.77) 32.27 (±4.58) 32.44 (±4.44) 

Birth weight: Mean, gr b 1876.94 (±813) 1856.42 (±715) 1836.06 (±852) 1906.37 (±839) 1908.91 (±849)

2. The proportion of babies receiving surfactants (all types) were more in babies > 1500 g.

Authors’ response: It should be noted that the data used in this research was extracted from a national system (Iranian Maternal and Neonatal Network (IMaN)). Therefore, the numerical values of the indicators used in this research are beyond the authority and control of the research team (the research team can only send data upon reasonable request). The data of this system is confidential and access to it is limited, and it can only be accessed with official permits for managerial-executive and research purposes, so there is no possibility of distorting this data. Based on the unpublished information of this department and interviews with the experts of this department, the verification of the data of this system has already been done with other executive purposes; Therefore, you can be sure of the correctness and accuracy of the data. Regarding the variable of birth weight of babies, we have only reported it descriptively.

In addition, according to Iran's national guideline for the administration of surfactant in neonates (under the title "National Guideline for Surfactant Prescription in Neonates"), the weight of neonates at birth is not a direct indicator for the administration or non-administration of surfactant. In this national guideline, there are four indicators for the prescription of surfactant, each indicator includes two variables, and when both variables are in specific and predetermined conditions, the neonatologist should prescribe surfactant, and the so-called surfactant prescription is considered reasonable (rational) in those conditions. These indicators for the administration of surfactant in neonates based on the national guidelines of Iran are as follows:

o Indicator 1: Premature neonates who need endotracheal intubation during postpartum resuscitation in the delivery /operating room.

o Indicator 2: Premature neonates reaching the stabilized health status in the delivery/operating room with nasal continuous positive airway pressure (NCPAP) and also need to increase continuous positive airway pressure (CPAP) to a maximum of 8 cm/H2O and FIO2 to more than 30% to 40% in order to maintain arterial oxygen saturation within an acceptable range.

o Indicator 3: Premature neonates showing typical respiratory distress syndrome (RDS)radiographic symptoms in the first 48 hours of life with a chest radiograph.

o Indicator 4: Premature or mature neonates with respiratory diseases who require endotracheal intubation.

Table 1. Summary of four indicators for surfactant prescription in infants, based on Iran's national guidelines

Prescription indicators Indicator variables Surfactant prescription should be done under the following conditions:

Indicator 1:

Variables

a and b Variable a: Gestational age ≤ 259 days (37weeks( 

 Variable b: Advancement in resuscitation operations in the operating /delivery room In need of resuscitation by intubation in the operating/delivery room

Indicator 2:

Variables

a and b Variable a: gestational age ≤ 259 days) 37weeks (

 Variable b: Type of Respiratory support before surfactant prescription and Status of the need for increasing CPAP and FIO2 NCPAP and need to increase CPAP > 8 cm/H2O or FIO2 > 30% 

Indicator3:

Variables

a and b Variable a: gestational age 

 Variable b: chest radiograph during the first 48 hours after birth ≤ 259 days )37 weeks( 

Indicator4:

Variables

a and b Variable a: Type of the respiratory distress disease Abnormal (with typical RDS radiographic signs)

 Variable b: Advancement in resuscitation operations RDS or MAS or PNA 

As you can see, although gestational age is one of the important variables in the three indicators for surfactant administration, birth weight is not directly mentioned in any of the indicators for surfactant prescription. It is true that the birth weight is affected by the gestational age, but according to the national guidelines of Iran, it is not directly used as an indicator for deciding on surfactant administration (we have attached the original version of the national guideline for the administration of surfactant in neonates, which was published by the Ministry of Health of Iran, in the additional files).

On the other hand, based on this guideline, babies less than 37 weeks of gestational age (provided they have the conditions mentioned in the guideline (Table 1)) are eligible to receive surfactant. And by examining the average weight of live babies born in the age group below 37 weeks, we determined that this average is over 1500 grams (average birth weight for neonates≤37 weeks=1686.75 grams). Therefore, although the mean weight of the babies receiving surfactant (all types) was over 1500 grams, their gestational age was less than 37 weeks; And according to Iran's national guidelines, gestational age is the criterion for prescribing surfactant, not birth weight. Below is the average weight for babies ≤37 weeks, separated by surfactant types (the research team can send the data file upon reasonable request).

Table 2. The average weight of newborns ≤ 37 weeks by surfactant type

Average birth weight for infants≤37 weeks Type of surfactant 

(Number of neonates≤37)

1670.728571 Alveofact (N =281)

1636.731771 BLES (N =1921)

1710.077087 Curosurf (N =7057)

1729.477675 Survanta (N =2598)

1686.753776 Total (N =11857)

In addition to all the explanations we provided above (that the criterion for prescribing surfactant is gestational age and not birth weight, and it was found that the group of infants who were eligible to receive surfactant in terms of age, their average weight was more than 1500 grams); In addition to these explanations, based on interviews with the directors of the Neonatal Health Department of the Iranian Ministry of Health and evidence, it was stated that...

In Iran's health system, the capabilities and facilities to provide care and keep premature babies alive in different regions are very different. In some areas of Iran (especially border provinces like Zahedan, etc.), the condition of NICUs is not suitable in terms of equipment and manpower, and premature babies and babies under 1500 grams (especially under 1200 grams) have very little chance of survival. For this reason and due to the limited number of beds and facilities in the NICU, in these areas, as an unwritten routine, more manpower and facilities are spent on babies with higher weight and older age because their chances of survival are higher.

On the other hand, in many of these areas, although they have a high reproduction rate, prenatal care is often very poor and the injection of antenatal corticosteroids is not done well. As a result, although it is expected that babies above 1500 do not have severe respiratory distress, in these areas even babies with higher weight and higher gestational age do not have good health conditions and may need surfactant injection. In general, in the border and deprived areas of Iran, which often have a high birth rate, babies under 1500 grams have very little chance of survival, and babies over 1500 grams have a high rate of respiratory distress, contrary to expectations. Therefore, this issue is one of the possible reasons for the higher amount of surfactant prescribed for babies above 1500 grams.

3. The average length of stay was quite short considering the GA and weight. Explanation?

Authors’ response: The findings of this study show that the average length of stay for the group of babies under 32 weeks who are naturally underweight (probably under 1500 gr) is 17.45 days. According to Table 3 in the manuscript, these values are: Alveofact=9.89, BLES=9.30, Curosurf 10.44, Survanta=10.24. On the other hand, the findings showed that the average length of stay for the group of babies over 32 weeks who are naturally heavier (probably over 1500 gr) is 9.96 days. According to Table 3 in the manuscript, these values are: Alveofact= 13.91, BLES= 17.27, Curosurf= 21.28, Survanta= 17.3. Therefore, with the decrease in the gestational age of babies (and most likely their weight loss), their average length of stay has obviously increased, which seems reasonable. 

There are other evidences from Iran similar to the results of our study (in terms of average length of stay considering birth weight and gestational age). For example:

o Example 1, Mousavi et al.'s study [3]: Comparison of the Efficacy of Three Natural Surfactants (Curosurf, Survanta, and Alveofact) in the Treatment of Respiratory Distress Syndrome Among Neonates. In this study, the average length of stay of neonates is 15.06 days (in our study, 13.71 days), while the average birth weight is 1839 grams (in our study, 1876.94 grams) and the average gestational age is 31.57 weeks (in our study, 32.24 weeks). Comparing the results of our study with this study shows that although the average length of stay in Mousavi's study was 1.38 days longer than in our study, the babies in this study had lower weight (37 gr) and lower gestational age (0.67 weeks) compared to our study.

Table 2. From the study of Mousavi et al.

Variable Type of Surfactant

 Survanta Alveofact Curosurf

Duration of hospital stay (mean ± SD) (days) 15.37 ± 14.2 15.47 ± 12.5 14.35 ± 12.6

Birth weight (mean ± SD) (gr) 1829 ± 782 1815 ± 729 1873 ± 859

Gestational age (mean ± SD) (weeks) 31.56 ± 3.8 31.46 ± 3.6 31.70 ± 3.8

o Example 2, Gharehbaghi et al.'s study [4]: Comparing the Efficacy of two Natural Surfactants, Curosurf and Alveofact, in Treatment of Respiratory Distress Syndrome in Preterm Infants. In this study, the average gestational age of the examined babies is 28.36 weeks (in our study, 32.24 weeks) and their average weight is 1316.50 grams (in our study, 1876.94 grams), while their average length of stay is 24.84 days (in our study, 13.71 days). It is clear that although in Gharehbaghi et al.'s study, the average length of stay of the infants was about 11 days longer than the infants in our study, but their age was about one month (3.88 weeks) less and their weight was 560 grams less than the infants in our study.

Table 1. From the study of Gharehbaghi et al.

Variable Type of Surfactant

 Curosurf group Alveofact group

Gestational age, wk 28.53±1.96 28.20±2.27

Birth weight, g 1350±555 1283±430

Mean duration of hospitalization 25.25±20.61 24.50±23.85

o Example 3, Najafian et al.'s study [5]: Comparison of efficacy and safety of two available natural surfactants in Iran, Curosurf and Survanta in treatment of neonatal respiratory distress syndrome. In this study, the average length of stay in the Survanta group infants is 15.36 days (in our study for infants in the Sorvanta group: 13.80 days), while their gestational age is 31.96 weeks (in our study for infants in the Sorvanta group: 32.44 weeks). From this comparison, it is clear that although the age of the babies in our study is only 0.48 weeks (equivalent to 3.36 days) more than the babies in Najafian et al.'s study, the average length of stay of the babies in our study is 1.56 days less than the babies in Najafian et al.'s study; these figures seem reasonable (more weight and less length of stay).

It should be noted that the data used in this research was extracted from a national system (Iranian Maternal and Neonatal Network (IMaN)). Therefore, the numerical values of the indicators used in this research are beyond the authority and control of the research team (the research team can only send data upon reasonable request). About the index of average length of stay, we selected this index based on similar studies, and we extracted the numerical values recorded for this index from the mentioned system and then analyzed it, without any change or manipulation. This national system is under the direct supervision of the Neonatal Health Department of the Ministry of Health of Iran. The data of this system is confidential and access to it is limited, and it can only be accessed with official permits for managerial-executive and research purposes, so there is no possibility of distorting this data. Based on the unpublished information of this department and interviews with the experts of this department, the verification of the data of this system has already been done with other executive purposes; Therefore, you can be sure of the correctness and accuracy of the data.

In a 40-minute online session, questions were asked by the first author and experts answered and provided guidance (some of the questions mentioned by the reviewers and editors in the manuscript review process). The interviewees were: Abbas Habibelahi and Parisa Mohagheghi as experts of neonatal department in ministry of health; Mohammad Heydarzadeh as the director of neonatal department in ministry of health. In addition to the fact that studies from Iran in terms of the average length of stay based on gestational age and birth weight were in line with the results of our study, policy makers in the field of newborn health also confirmed these figures. Based on the interview with them, they stated that due to the lack of hospital beds, especially for babies in Iran, the overall effort is to reduce the length of stay as much as possible so that people in need can be hospitalized faster. It seems that such figures about the average length of stay according to weight and gestational age are common in Iran.

4. DALY/YLL/YLD should be expressed in years.

Authors’ response: Thank you very much for your valuable comment. According to your suggestion, we also expressed the values of DALY/YLL/YLD in the form of years. Please check this revision in Table 2. Although DALY/YLL/YLD values have been reported and calculated based on the day in order to increase the accuracy of calculations in the decision matrix, these values are also written in the form of years in Table 2) according to your comment). These are as follows:

DALY, day (year) 14820.53(40.6041) 17909.77(49.0681) 14927.34(40.8968) 14983.65(41.0510) 13939.82(38.1912)

YLL, day (year) 14799.98(40.5478) 17893.47(49.0232) 14909.05(40.8467) 14961.81(40.9912) 13920.79(38.1391)

YLD, day (year) 20.57(0.0563) 16.41(0.0449) 18.32(0.0501) 21.86(0.0598) 19.03(0.0521)

5. Can the course of babies included in studies be compared? Number of babies needing invasive/non invasive ventilation, shock, IVH, PVL, etc. These all factors can also have bearing on the indicators.

Authors’ response: Yes, this comparison is possible, we examined a large number of variables before conducting the main research analyzes. The research team, from the beginning of this research, based on the comprehensive data set that was extracted from the Iranian Maternal and Neonatal Network (IMAN net), examined the groups of infants in terms of many characteristics that their data is available in the system, and it was found that there is no significant difference in these characteristics. But due to the large number of these variables, it is challenging to present them in the text of the manuscript, express the results and discuss them; and the research team prefers to report only common variables used in other similar studies. However, based on this comment we made extensive corrections, and provided a lot of information. Please pay attention to the following explanations:

In Table 1 of the manuscript, we only stated five very common characteristics that were reported by other similar studies, to avoid data overload in the manuscript; the information in this table briefly shows the similarity of four groups of infants based on baseline parameters. And in Table 2, to perform the main analysis of the manuscript (MABAC and CRITIC), we selected the variables that were firstly commonly used by other researchers and secondly those variables had a statistically significant difference between the four groups. In fact, the variables that were included in the multi-criteria decision analysis (MABAC and CRITIC method) were definitely statistically significantly different between the four groups of infants. And about other variables that were not significantly different between groups of infants, these variables were never included in the main analyzes (MABAC and CRITIC method) of the manuscript at all. 

Therefore, if a variable was related but was not included in multi-criteria decision analysis, either its data was not available or it was not statistically different between infants. For example, regarding the number of infants who need invasive/non-invasive ventilation, the index of non-invasive ventilation was not significantly different between the four groups of infants, but the index of invasive ventilation was significantly different. Therefore, the invasive ventilation index was considered in the main analyzes of the research (MABAC and CRITIC method); While non-invasive ventilation was not considered. In the text of the manuscript, mechanical ventilation means invasive mechanical ventilation. In response to your comment, we have corrected this term throughout the manuscript (Invasive mechanical ventilation replaced mechanical ventilation). 

Based on your feedback, we added IVH, PVL and a large number of other variables that had the potential to directly and indirectly affect the research results to the appendices (but based on the available data, these variables were not significantly different between the babies of the four groups). We provided a descriptive report of these variables and added Table S3.1 in S3_File. And we mentioned them briefly in the text of the manuscript. As follows:

“And it was determined in advance that before the administration of surfactant, there was no statistically significant difference in the health status of infants in different surfactant groups (see some other variables in Table S3.1 in S3_File).”

"The indicators investigated in this research were extracted from the literature on this topic. The indicators that were included in the main analyzes of the article were previously determined to be statistically significantly different between the four groups of infants. Although there were some other relevant indicators, they were removed from the analysis due to the lack of statistically significant differences (see Table S3.1 in S3_File). The selected indicators are as follows:”

 By expressing this comment, you have provided us with this opportunity to publish our extensive collection of data and analysis. Thank you again. 

It should be noted that in expressing the results and discussion in the text of the manuscript, we strongly emphasized everywhere that the ranking done in the types of surfactants is based on a number of selected indicators. And finally, although based on the comprehensive data set we had and by referring to similar studies, the most important variables (indicators) related to the subject of this research were considered, nevertheless, it may be a variable whose data was not accessible to the research team. For example, about the shock variable that you mentioned, there was no data recorded in the national network of Iran. According to your comment, we stated this issue as one of the limitations of the research and we hope it will be acceptable. We stated this limitation as follows:

“In this research, we were limited to the data extracted from the Iranian Maternal and Neonatal Network (IMAN net), and it was not possible to comprehensively examine all the factors.”

If you have a specific suggestion or recommendation about this comment, please share it with us; If we can, we will implement your suggestion.

Discussion

1. The evidence on efficacy of surfactants is already there so the studies which have looked at surfactant in a similar perspective (economic efficacy).

Authors’ response: According to the research team's search, in all the studies conducted to compare the types of surfactants, economic efficiency has been investigated along with clinical efficiency, and we did not find a study that only investigated economic efficiency. The studies presented in the section "4.2.3 Direct medical cost (I3)" have also often been conducted with the aim of evaluating the economic-clinical efficiency. According to your comment, we wrote the purpose of the mentioned studies in the section "4.2.3 Direct medical cost (I3)". Please check these changes in the main text of the manuscript (Sarokolai et al: to determine the most efficacious type of surfactant,; Zayek et al: to compare the efficacy and safety of Calfactant and Curosurf; Marsh et al: to compare the pharmacoeconomic profiles of Survanta and Curosurf via a cost-minimization analysis; Brown et al: a study aimed at evaluating economic and therapeutic efficiency; Another study aimed at comparing the efficacy and safety of bovine lung phospholipid and Poractant Alfa; A cost-effectiveness analysis study of surfactant therapy in the treatment of NRDS ).

In addition, in response to your comment, we added other studies that were conducted with the aim of examining the economic-clinical efficiency (including studies: Brown et al.'s study: A cost-effectiveness analysis study; Another study aimed at comparing the efficacy and safety of bovine lung phospholipid and Poractant Alfa). Please check the revisions related to this comment in the "4.2.3 Direct medical cost (I3)" section. These are as follows:

“4.2.3 Direct medical cost (I3)

The results of this study showed that BLES and Survanta, as the superior surfactants, imposed significantly lower costs on the health system, compared to Alveofact and Curosurf. Sarokolai et al. examined the cost index in preterm neonates with RDS to determine the most efficacious type of surfactant, showing that this index was significantly lower in the group treated with BLES than the one receiving Curosurf. As such, in 91.50% of infants who received Curosurf, cost exceeded $200 (inflation-adjusted cost = $ 273.48), although that cost was only observed in case of 8.50% of infants who used BLES [40]. In a retrospective cohort study of infants at risk of RDS to compare the efficacy and safety of Calfactant and Curosurf, Zayek et al. found that the cost of Curosurf-based treatment per patient was $ 1,160.62 (inflation-adjusted cost: $1229.56), which was 38% higher than the cost imposed by using Calfactant ($838.34 (inflation-adjusted cost: $877.08)) [16]. Marsh et al conducted a study to compare the pharmacoeconomic profiles of Survanta and Curosurf via a cost-minimization analysis. These analyses would suggest Curosurf may offer a less costly, clinically-equivalent option. Different treatment models using Curosurf (compared to Survanta) resulted in cost savings ranging from 53% ($949.67 (inflation-adjusted cost: $1296.21)) to 20% ($180 (inflation-adjusted cost: $245.68)) [46]. In a study aimed at evaluating economic and therapeutic efficiency (based on drug therapy cost index, duration of respiratory support, duration of hospitalization, side effects, etc.), Brown et al. reported higher average medication costs ($1756.44 vs. $1329.78 (inflation-adjusted cost: $1860.77 vs. $1408.77)) for Poractant Alfa (Curosurf) compared with Beractant (Survanta). While many clinical indicators were not significantly different between the groups [47]. A cost-effectiveness analysis study of surfactant therapy in the treatment of NRDS showed that the use of Poractant Alfa (Curosurf) is a superior option compared to Beractant (Survanta). Cost-effectiveness ratio was €4585 ($5067.57) per saved life for Poractant Alfa and €5087 ($5590.35) per saved life for Beractant [48]. Another study aimed at comparing the efficacy and safety of bovine lung phospholipid and Poractant Alfa injection in the treatment of neonatal hyaline membrane disease showed that treatment costs in the for Poractant Alfa group were significantly lower than the bovine lung phospholipid group [49].”

2. I am not sure how the cost can be compared from different countries at different time points. Cost is variable and dependent on plethora of factors.

Authors’ response: Other evidence including Sarokolai et al [6], Marsh et al [7], Zayek et al [8], Yuan et al [9], Yagudina et al [10], and Brown et al [11] have also used the cost index to compare the efficacy and pharmacoeconomics of different surfactants.

In all the studies reported in the "4.2.3 Direct medical cost (I3)" section, the price of surfactant (according to the type of surfactant) and the number of prescription doses are considered as one of the most important cost components. Therefore, comparing these studies based on the type of cost is correct and possible. (For example, Sarokolai et al.'s study: surfactant vial cost (BLES recipient paid less because BLES was cheaper than Curosurf); Marsh et al.'s study: surfactant vial cost and number of doses (the only cost to be compared between drugs for CMA considered, the cost of the initial and subsequent doses); Zayek et al.'s study: surfactant vial cost and its different doses (costs per patient are determined by the price difference between the 2 products and the average number of doses per patient); etc.)

On the other hand, as you mentioned, since the mentioned costs are related to different countries and different periods of time, therefore, their direct comparison is not correct. As a result, based on your comment, we converted the costs mentioned in different studies into a single currency unit (dollars), and then adjusted them according to the inflation rate of the country under study (Since the data of 2018 was used for the analysis of this study and the data of the most recent study compared with this research was also for 2018, therefore the adjustment was made according to the inflation rate until 2018). We made this edit in the form of "(inflation-adjusted cost: $)", and we used the World Data site (https://www.worlddata.info/inflation.php ) to determine the inflation rate of the countries. Thank you very much for this valuable comment, these corrections will definitely improve the quality of the manuscript. Please check the revisions related to this comment in the "4.2.3 Direct medical cost (I3)" section. These are as follows:

“4.2.3 Direct medical cost (I3)

The results of this study showed that BLES and Survanta, as the superior surfactants, imposed significantly lower costs on the health system, compared to Alveofact and Curosurf. Sarokolai et al. examined the cost index in preterm neonates with RDS to determine the most efficacious type of surfactant, showing that this index was significantly lower in the group treated with BLES than the one receiving Curosurf. As such, in 91.50% of infants who received Curosurf, cost exceeded $200 (inflation-adjusted cost = $ 273.48), although that cost was only observed in case of 8.50% of infants who used BLES [40]. In a retrospective cohort study of infants at risk of RDS to compare the efficacy and safety of Calfactant and Curosurf, Zayek et al. found that the cost of Curosurf-based treatment per patient was $ 1,160.62 (inflation-adjusted cost: $1229.56), which was 38% higher than the cost imposed by using Calfactant ($838.34 (inflation-adjusted cost: $877.08)) [16]. Marsh et al conducted a study to compare the pharmacoeconomic profiles of Survanta and Curosurf via a cost-minimization analysis. These analyses would suggest Curosurf may offer a less costly, clinically-equivalent option. Different treatment models using Curosurf (compared to Survanta) resulted in cost savings ranging from 53% ($949.67 (inflation-adjusted cost: $1296.21)) to 20% ($180 (inflation-adjusted cost: $245.68)) [46]. In a study aimed at evaluating economic and therapeutic efficiency (based on drug therapy cost index, duration of respiratory support, duration of hospitalization, side effects, etc.), Brown et al. reported higher average medication costs ($1756.44 vs. $1329.78 (inflation-adjusted cost: $1860.77 vs. $1408.77)) for Poractant Alfa (Curosurf) compared with Beractant (Survanta). While many clinical indicators were not significantly different between the groups [47]. A cost-effectiveness analysis study of surfactant therapy in the treatment of NRDS showed that the use of Poractant Alfa (Curosurf) is a superior option compared to Beractant (Survanta). Cost-effectiveness ratio was €4585 ($5067.57) per saved life for Poractant Alfa and €5087 ($5590.35) per saved life for Beractant [48]. Another study aimed at comparing the efficacy and safety of bovine lung phospholipid and Poractant Alfa injection in the treatment of neonatal hyaline membrane disease showed that treatment costs in the for Poractant Alfa group were significantly lower than the bovine lung phospholipid group [49].”

3. The need for mechanical ventilation can be due to multiple reasons or other single etiologic factors than RDS. 

Authors’ response: Thank you very much for this comment. Yes, you are right, the need for mechanical ventilation may be due to reasons other than RDS. But it should be noted that the target population of this research is infants who were firstly diagnosed with RDS, and secondly, surfactant was prescribed to them. And, as evidence shows, one of the short-term effects of surfactant injection is reducing the need for mechanical ventilation [12]. And in the population studied in this research, like other similar studies, the need for mechanical ventilation after surfactant injection can indicate the effectiveness of the surfactant. Mousavi et al.'s study [3], and Mirzarahimi et al.'s study [12], which had the same target population and research objectives as our research, have used this index (need for mechanical ventilation) as an index to measure the effectiveness of various surfactants (a summary of these studies is provided below). 

Some of the evidence reviewed to respond to this comment:

Mousavi et al.'s study [3]:

o Title: Comparison of the Efficacy of Three Natural Surfactants (Curosurf, Survanta, and Alveofact) in the Treatment of Respiratory Distress Syndrome Among Neonates

o Clinical parameters for comparing three surfactants: hospital-stay length, mechanical ventilation requirement, and...

o Result: InSurE failure and mechanical ventilation support requirement in neonates over 32 weeks was significantly lower in the Survanta group (P = 0.019). And ....

Mirzarahimi et al.'s study [12]:

o Title: Comparison efficacy of Curosurf and Survanta in preterm infants with respiratory distress syndrome

o Clinical parameters for comparing two surfactants: hospital-stay length, need for ventilation, repeated doses, mortality rate, and...

o Result: …but the need for repeated doses in Curosurf group and need for ventilation in Survanta group is less than others/ … and in mean duration of ventilation Survanta group with 8 days was lower than Curosurf group with 10.5 days [P=0.001].

Dani et al.'s study [13]:

o Title: Analysis of the cost-effectiveness of surfactant treatment (Curosurf) in respiratory distress syndrome therapy in preterm infants: early treatment compared to late treatment

o Parameters for comparing three surfactants: The duration of the need for mechanical ventilation, and Variation in Mechanical Ventilation cost

o Result: The cost of treatment with surfactant was greater in the early group, but this was compensated by the greater cost of treatment with Mechanical Ventilation (MV) in the late group.

In addition, as shown in Table 1 of the manuscript, infants in the four surfactant groups do not differ significantly based on baseline characteristics. In Table 1 of the manuscript, we only stated five very common characteristics that were reported by other similar studies, to avoid data overload in the manuscript. While the research team, from the beginning of this research, based on the comprehensive data set that was extracted from the Iranian Maternal and Neonatal Network (IMAN net), examined the groups of infants in terms of many characteristics that their data is available in the system, and it was found that there is no significant difference in these characteristics. But due to the large number of these variables, it is challenging to present them in the text of the manuscript, express the results and discuss them; and the research team prefers to report only common variables used in other similar studies. In fact, with the aim of ensuring that the babies of the four groups were similar before the administration of surfactant, we examined a large number of variables that directly and indirectly affected the subject of the research, and finally it was found that most of these variables had no significant difference between the four groups (variables related to the clinical conditions of the newborn, risk factors related to the mother, and diseases and abnormalities of newborns, etc.). Many of these variables directly and indirectly affect the "need for mechanical ventilation" index. Considering the large number of investigated variables, therefore, it can probably be concluded that many reasons or factors other than RDS, which may affect the "need for mechanical ventilation" index, are probably not significantly different between the four investigated groups. However, we also mention this issue as one of the limitations of the research. Additionally, based on your comments, we provided a descriptive report of these variables and added Table S3.1 in S3_File. And we mentioned them briefly in the text of the manuscript. By expressing this comment, you have provided us with this opportunity to publish our extensive collection of data and analysis. Thank you again.

It should be noted that although some variables that affect the need for mechanical ventilation have been reported, it is still not possible to comprehensively examine all factors and we are limited to the data recorded in the national system. Based on your comment, we stated this issue as one of the limitations of the research in the manuscript. We thank you again for this valuable comment. If you have specific advice in this regard, please share it with us and we will do it if we can. Please check these revisions in the last paragraph of the discussion section. These revisions are as follows:

“In addition, another limitation of this research is that although according to the evidence, the need for mechanical ventilation and its duration is one of the indications of the efficacy of surfactant therapy, it is possible that factors other than RDS caused it. In this research, we were limited to the data extracted from the Iranian Maternal and Neonatal Network (IMAN net), and it was not possible to comprehensively examine all the factors. Another main limitation…”

 

References:

1. Diakoulaki, D., G. Mavrotas, and L. Papayannakis, Determining objective weights in multiple criteria problems: The critic method. Computers & Operations Research, 1995. 22(7): p. 763-770.

2. Pamučar, D. and G. Ćirović, The selection of transport and handling resources in logistics centers using Multi-Attributive Border Approximation area Comparison (MABAC). Expert systems with applications, 2015. 42(6): p. 3016-3028.

3. Mussavi, M., K. Mirnia, and K. Asadollahi, Comparison of the efficacy of three natural surfactants (Curosurf, Survanta, and Alveofact) in the treatment of respiratory distress syndrome among neonates: a randomized controlled trial. Iranian Journal of Pediatrics, 2016. 26(5).

4. Gharehbaghi, M.M. and S. Yasrebi, Comparing the efficacy of two natural surfactants, Curosurf and Alveofact, in treatment of respiratory distress syndrome in preterm infants. International Journal of Women’s Health and Reproduction Sciences, 2014. 2(4): p. 245-8.

5. Najafian, B., et al., Comparison of efficacy and safety of two available natural surfactants in Iran, Curosurf and Survanta in treatment of neonatal respiratory distress syndrome: A randomized clinical trial. Contemporary clinical trials communications, 2016. 3: p. 55-59.

6. Sarokolai, Z.K., et al., BLES versus curosurf for treatment of respiratory distress in preterm neonates and their adverse effects. Iranian Journal of Pediatrics, 2018. 28(4).

7. Marsh, W., et al., A cost minimization comparison of two surfactants—beractant and poractant alfa—based upon prospectively designed, comparative clinical trial data. The Journal of Pediatric Pharmacology and Therapeutics, 2004. 9(2): p. 117-125.

8. Zayek, M.M., F.G. Eyal, and R.C. Smith, Comparison of the pharmacoeconomics of calfactant and poractant alfa in surfactant replacement therapy. The Journal of Pediatric Pharmacology and Therapeutics, 2018. 23(2): p. 146-151.

9. Yuan, G., et al., Comparison of the efficacy and safety of bovine lung phospholipid and poractant alfa injections in the treatment of neonatal hyaline membrane disease with continuous positive airway pressure. Int J Clin Exp Med, 2019. 12(3): p. 3007-3013.

10. Yagudina, R., A. Kulikov, and V. Serpik, PRS50-cost-effectiveness analysis of surfactant therapy for the treatment of respiratory distress syndrome newborn in the Russian Federation. Value in Health, 2018. 21: p. S412.

11. Brown, S., J. Hurren, and H. Sartori, Poractant alfa versus beractant for neonatal respiratory distress syndrome: a retrospective cost analysis. The Journal of Pediatric Pharmacology and Therapeutics, 2018. 23(5): p. 367-371.

12. Mirzarahimi, M. and M. Barak, Comparison efficacy of Curosurf and Survanta in preterm infants with respiratory distress syndrome. Pak J Pharm Sci, 2018. 31(2): p. 469-472.

13. Dani, C., et al., Analysis of the cost-effectiveness of surfactant treatment (Curosurf®) in respiratory distress syndrome therapy in preterm infants: early treatment compared to late treatment. Italian journal of pediatrics, 2014. 40(1): p. 1-7.

---

## [Editor Report · Decision Letter 1]

29 May 2023

Comparing the Clinical and Economic Efficiency of Four Natural Surfactants in Treating Infants with Respiratory Distress Syndrome

PONE-D-23-01167R1

Dear Dr. Shojaei,

We’re pleased to inform you that your manuscript has been judged scientifically suitable for publication and will be formally accepted for publication once it meets all outstanding technical requirements.

Kind regards,

Stefan Grosek, Ph.D., M.D.,

Academic Editor

PLOS ONE

Additional Editor Comments (optional):

Dear Payam Shojaei and co-authors

Thank you for your excellent revision of your article and suuported informations, explanation, and corrections received. Yourr design, statististical approach which support the Result nad Discussion section is very inovative and intersting, well written and supported with relevant literature.
---

## [Editor Report · Acceptance letter]

22 Jun 2023

PONE-D-23-01167R1 

Comparing the clinical and economic efficiency of four natural surfactants in treating infants with respiratory distress syndrome 

Dear Dr. Shojaei:

I'm pleased to inform you that your manuscript has been deemed suitable for publication in PLOS ONE. Congratulations! Your manuscript is now with our production department. 

Kind regards, 

on behalf of

Professor Stefan Grosek 

Academic Editor

PLOS ONE